# Smoothing DiLoCo with Primal Averaging for Faster Training of LLMs

## Abstract

We propose Generalized Primal Averaging (GPA), an extension of Nesterov's method in its primal averaging formulation that addresses key limitations of recent averaging-based optimizers such as DiLoCo and Schedule-Free (SF) in the non-distributed setting. These two recent algorithmic approaches improve the performance of base optimizers such as AdamW through different iterate averaging strategies. Schedule-Free explicitly averages iterates at every step, while DiLoCo performs implicit averaging by periodically aggregating trajectories, called pseudo-gradients, to update the model parameters. This periodic averaging introduces a two-loop structure, increasing its memory requirements and the number of hyperparameters to tune. To address these limitations, GPA smoothens DiLoCo by averaging iterates at every iteration using two interpolation constants. When applied to language model pre-training, GPA consistently outperforms DiLoCo while removing the two-loop structure, simplifying hyperparameter tuning and reducing memory overhead to a single additional buffer. Furthermore, we prove that for any base optimizer with regret bounded by $\mathcal{O}(\sqrt{T})$, where $T$ is the number of iterations, GPA can match or exceed the convergence guarantee of the original optimizer, depending on the choice of the interpolation constants.

## 1 Introduction

As large language models (LLMs) demonstrate increasingly remarkable capabilities at scale (Achiam et al., 2023; Llama Team, 2024; Liu et al., 2024a), the pre-training phase has become one of the most expensive stages in the language model training pipeline, often costing hundreds of millions of dollars per run. This significant investment has driven the development of training algorithms and optimizers that enhance the efficiency, scalability, and robustness of language model pre-training. One significant area of research is the design of training algorithms for scalable distributed learning. Among these, the DiLoCo algorithm has emerged as the leading practical approach (Douillard et al., 2023; Liu et al., 2024b; Douillard et al., 2025; Charles et al., 2025).

DiLoCo notably outperforms AdamW, even in non-distributed setups, due to its novel combination of the Nesterov optimizer with the Lookahead method, also called Step-$K$ Nesterov (Zhang et al., 2019; Kallusky et al., 2025). The method computes a trajectory that accumulates multiple updates from a base optimizer on an inner set of weights, called the *pseudo-gradient*, applies Nesterov momentum on the pseudo-gradients to update an outer set of weights, then resets the inner set of weights to the current outer weights. In a non-distributed setup, DiLoCo delivers substantial efficiency gains; for instance, when applied to AdamW on a 160 million parameter language model, this approach yields speedups up to 34.78%; see Figure 1b.

A particularly intriguing behavior of DiLoCo is that its performance improves as the number of inner steps increases. With each base optimizer step, DiLoCo's outer weights drift farther from its inner weights, similar to meta-learning optimizers such as Reptile (Nichol & Schulman, 2018) and First-Order MAML (Finn et al., 2017). As a result, updates to the outer weights occur only at periodic intervals, causing information from the data to be integrated in a discontinuous, choppy manner rather than smoothly at every iteration. This restriction on information flow to the outer weights appears unnecessary from an optimization perspective, yet counterintuitively improves its performance; see Figure 1a.

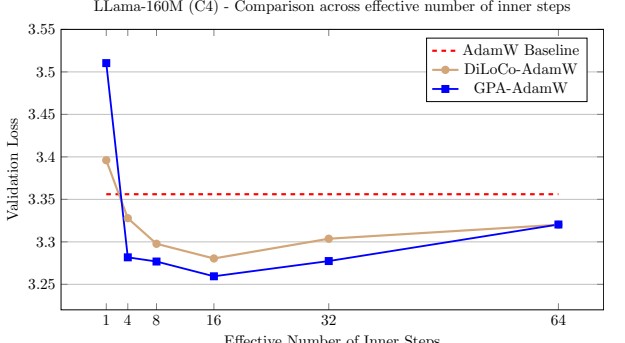 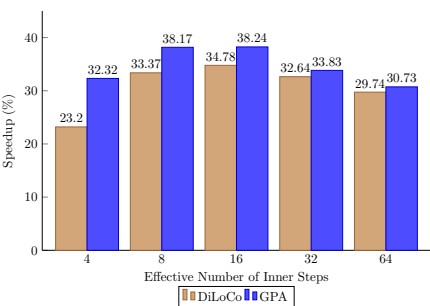

(a) Both GPA and DiLoCo using AdamW as their base optimizer significantly outperform a strong AdamW baseline for training a 160M parameter Llama model. Notably, increasing the number of inner steps (up to 16) improves the performance of DiLoCo. Unlike DiLoCo, GPA updates the parameters at every step, but uses a heuristic to choose its interpolation constants to match the number of inner steps for DiLoCo.

(b) Speedup achieved by DiLoCo and GPA in reducing the number of steps to reach AdamW's final validation loss, across different effective numbers of inner steps. GPA and DiLoCo attain the highest speedup of 38.24% and 34.78% respectively for the same interval of 16.

Figure 1: Comparison of validation loss and speedup for AdamW, DiLoCo, and GPA.

Concurrently, the Schedule-Free optimizer recently won the AlgoPerf Algorithmic Efficiency challenge self-tuning track (Dahl et al., 2023; Defazio et al., 2024). Its core novelty lies in computing gradients at a point that interpolates between the uniform average of past weights and the current weights. Empirically, Schedule-Free matches the performance obtained by using learning rate schedules without using any schedule explicitly, while providing stronger theoretical last-iterate convergence guarantees similar to Polyak-Ruppert averaging (Ruppert, 1988; Polyak, 1990; Polyak & Juditsky, 1992).

In this paper, we argue that these two lines of work – DiLoCo and Schedule-Free – are closely related and can be generalized and improved through a unified framework of *primal averaging*. Specifically, our contributions are as follows:

- We propose a generalization of Nesterov's method in its primal averaging formulation called *Generalized Primal Averaging* (GPA), which smooths DiLoCo by incrementally averaging iterates at every step.

- In contrast to DiLoCo, GPA eliminates the two-loop structure, thereby requiring only a single additional buffer with less hyperparameters to tune. The method also demonstrates more stable training behavior than DiLoCo.

- Our experiments demonstrate that GPA consistently outperforms non-distributed DiLoCo and AdamW on dense 160 million and 1 billion parameter language models. This is further validated on the ImageNet ViT workload.

- We also provide a theoretical justification for GPA through convergence guarantees that demonstrate improved convergence over the base optimizer under some circumstances.

## 2 BACKGROUND

We frame language model pre-training as the expected risk minimization problem

$$\min_{x \in \mathbb{R}^n} F(x) = \mathbb{E}_{\xi \sim \mathcal{D}} \left[ f(x; \xi) \right], \tag{1}$$

where $\xi \sim \mathcal{D}$ is drawn from an underlying stationary data distribution $\mathcal{D}$. We assume that each optimizer step has access to the stochastic minibatch gradient $g(x^{(t)}; \xi^{(t)}) \in \partial f(x^{(t)}; \xi^{(t)})$ evaluated at each iteration $t$ on a minibatch of data $\xi^{(t)}$, over a total of $T$ steps.[1]

---

[1]We assume that $f$ is convex for the convergence analysis, but we verify its performance on non-convex, possibly non-smooth functions.

We also assume that the base optimizer is of the form $x^{(t+1)} = x^{(t)} + \gamma^{(t)} d^{(t)}$ with learning rate $\gamma^{(t)} > 0$ and search direction $d^{(t)} \in \mathbb{R}^n$. The search direction is most commonly defined as $d^{(t)} = -H^{(t)} m^{(t)}$, where $m^{(t)} \in \mathbb{R}^n$ is a gradient estimator, and $H^{(t)} \in \mathbb{R}^{n \times n}$ is a symmetric positive definite preconditioner matrix. This includes popular methods such as SGD, Adam, Shampoo, SOAP, AdEMAMix, or Muon for different choices of $m^{(t)}$ and $H^{(t)}$ (Robbins & Monro, 1951; Kingma & Ba, 2014; Gupta et al., 2018; Loshchilov & Hutter, 2019; Anil et al., 2020; Shi et al., 2023; Vyas et al., 2024; Jordan et al., 2024; Pagliardini et al., 2025; Eschenhagen et al., 2025).

## 2.1 DIFFERENT FORMULATIONS OF NESTEROV MOMENTUM

Nesterov momentum has played a critical role in optimization for deep learning (Sutskever et al., 2013). Despite its importance, there is still substantial confusion in the literature regarding Nesterov's formulation, as it can be written in at least seven different ways (Defazio, 2019). These formulations are equivalent in the sense that a direct mapping exists between them, but they may not return the same iterate.

For instance, Nesterov's method was popularized for deep learning in *Sutskever's formulation* (Sutskever et al., 2013), which presents the algorithm as:

$$\begin{aligned}
b^{(t)} &= \mu b^{(t-1)} - \gamma^{(t)} g(x^{(t)} + \mu b^{(t-1)}; \xi^{(t)}), \\
x^{(t+1)} &= x^{(t)} + b^{(t)},
\end{aligned} \tag{2}$$

where $\mu > 0$ is the momentum hyperparameter and $b^{(t)} \in \mathbb{R}^n$ is the momentum buffer initialized at $b^{(0)} = 0$. An alternative formulation, which we call the *modern formulation*, is used by software libraries such as PyTorch[2] and JAX[3] due to its ease of use:

$$\begin{aligned}
b^{(t)} &= \mu b^{(t-1)} + g(x^{(t)}; \xi^{(t)}), \\
x^{(t+1)} &= x^{(t)} - \gamma^{(t)} [\mu b^{(t)} + g(x^{(t)}; \xi^{(t)})].
\end{aligned} \tag{3}$$

In both formulations, we maintain a momentum buffer that averages the gradients seen throughout the training process. However, unlike Sutskever's formulation (equation 2), the modern formulation (equation 3) uses the iterate $x^{(t)}$ directly for the gradient computation, rather than the ancillary point $x^{(t)} + \mu b^{(t-1)}$, simplifying its practical implementation. If both formulations are run side-by-side with the same seed, they will evaluate gradients at exactly the same points, but their validation losses at iterates $x^{(t)}$ for each method will differ.

Our approach instead builds upon a third form, which we call the *primal averaging formulation*:

$$\begin{aligned}
y^{(t)} &= \mu x^{(t)} + (1 - \mu) z^{(t)}, \\
z^{(t+1)} &= z^{(t)} - \gamma^{(t)} g(y^{(t)}; \xi^{(t)}), \\
x^{(t+1)} &= \mu x^{(t)} + (1 - \mu) z^{(t+1)},
\end{aligned} \tag{4}$$

with $\mu \in [0, 1)$. The first mention of this three-sequence form that we are aware of is by Lan (2012), although it was only studied under a time-varying $\mu$.

Unlike the Sutskever and modern formulations framed in equations 2 and 3, the primal averaging formulation in equation 4 explicitly names two iterate sequences: a sequence where the gradients (or, more generally, the search directions) are computed at, i.e., the *gradient computation sequence* $\{y^{(t)}\}_{t=1}^T$, as well as another sequence used for model evaluation that accumulates a running average of updated iterates $\{z^{(t)}\}_{t=1}^T$, i.e., the *model evaluation sequence* $\{x^{(t)}\}_{t=1}^T$. Since $y^{(t)}$ interpolates the smoothed sequence $x^{(t)}$ and unsmoothed sequence $z^{(t)}$, it increases the contribution of the gradient update to $y^{(t)}$ compared to $x^{(t)}$. This explicit formulation is convenient for implementation and theoretical analysis, and naturally leads to a view of acceleration as built upon *iterate averaging*, rather than from the physics-inspired intuition of *gradient averaging* behind momentum that is more commonly introduced.

We summarize the relationship between the modern and primal averaging formulations in Proposition 1 below.

---

[2]https://docs.pytorch.org/docs/2.8/generated/torch.optim.SGD.html
[3]https://optax.readthedocs.io/en/latest/api/optimizers.html#optax.sgd

**Proposition 1.** *Given fixed learning rates $\gamma_{\text{primal}}, \gamma_{\text{modern}} > 0$, Nesterov's primal averaging formulation (equation 4) is equivalent to Nesterov's modern formulation (equation 3) in the sense that*

$$y_{\text{primal}}^{(t)} = x_{\text{modern}}^{(t)} \quad and \quad b_{\text{modern}}^{(t)} = \frac{1}{(1-\mu)\,\gamma_{\text{primal}}} \left( x_{\text{primal}}^{(t)} - x_{\text{primal}}^{(t+1)} \right), \tag{5}$$

*when $\mu_{\text{primal}} = \mu_{\text{modern}} = \mu$ and $(1-\mu)\,\gamma_{\text{primal}} = \gamma_{\text{modern}}$.*

The proof of this simple statement is rather technical, so we defer it to Appendix D. Similar formulations and equivalences can be derived for Polyak momentum (Polyak, 1964; Defazio, 2020; Ziyin et al., 2020); see Appendix B.

**Remark.** It is important to acknowledge that the equivalence between the primal averaging and modern formulations of Nesterov momentum holds only when the learning rates are *constant*. When learning rate schedules are introduced, achieving this equivalence would require the momentum parameter to vary with each iteration. Furthermore, the restriction on the choice of $\mu$ differs between the modern and primal averaging formulations. These different interpretations based on *gradient averaging* versus *iterate averaging* produce differing perspectives for hyperparameter tuning, which can have a significant impact on the algorithm's practical performance.

## 2.2 Non-Distributed DiLoCo and its Weaknesses

DiLoCo was originally introduced as a distributed algorithm for cross-datacenter training (Douillard et al., 2023). In the non-distributed setup, it computes multiple inner steps of the base optimizer on the *inner weights*, then applies Nesterov (equation 3) on the *pseudo-gradient*, the difference between the previous and updated inner model weights, to the *outer weights*. The inner weights are then reset to the outer weights.

DiLoCo requires storing two additional optimizer states of the same shape as the model parameters: the momentum buffer $b^{(t)}$ and the current model parameters $x^{(t)}$ (also known as the *outer weights*). DiLoCo's handling of *fast* inner weights and *slow* outer weights can be interpreted as a modified Lookahead method that applies Nesterov momentum to the outer weight updates (Zhang et al., 2019). The method was recently analyzed in Khaled et al. (2025), and demonstrated significant compute factor gains in the non-distributed setting in Kallusky et al. (2025).

A simplified version of non-distributed DiLoCo with $H$ inner steps of the base optimizer can be described as:

$$\begin{aligned} p^{(t)} &= x^{(t)} - \texttt{BaseOptIteration}(x^{(t)}; \{\gamma^{(j)}\}_{j=1}^{H}, H) \\ b^{(t)} &= \mu b^{(t-1)} + p^{(t)} \\ x^{(t+1)} &= x^{(t)} - \tilde{\gamma}[\mu b^{(t)} + p^{(t)}], \end{aligned} \tag{6}$$

where $\tilde{\gamma} > 0$ is the outer learning rate and $\texttt{BaseOptIteration}$ applies $H$ iterations of the base optimizer to the iterate $x^{(t)}$ with inner learning rates $\{\gamma^{(j)}\}_{j=1}^{H}$. While DiLoCo originally introduced AdamW as the base optimizer, DiLoCo has been generalized to other optimizers such as Muon (Thérien et al., 2025). A complete description of the algorithm is provided in Appendix C. As noted in Kallusky et al. (2025), applying Nesterov on the pseudo-gradient with multiple base optimizer steps is capable of surpassing the performance of the base optimizer alone, which explains DiLoCo's ability to match the synchronous baseline, such as AdamW, in the multi-worker setting.

**Weaknesses in DiLoCo's hierarchical framework.** However, this two-level structure is undesirable. From an *algorithmic perspective*, one would prefer to average iterates on-the-fly, as opposed to averaging trajectories that implicitly contain multiple iterations of the base optimizer. From the *users' perspective*, the two-level structure introduces an additional copy of the model weights required to compute the pseudo-gradient, and introduces additional hyperparameters to tune, e.g., the inner and outer learning rates, momentum, and number of inner steps. Lastly, from the *distributed training perspective*, DiLoCo couples the number of inner steps as a hyperparameter for both local SGD as well as for the modified Nesterov algorithm, causing the algorithm's performance to counterintuitively improve as the number of base optimizer steps increases. One would instead expect that communicating more often should always be beneficial. These challenges motivate the development of a new algorithm that *removes the two-level structure* while offering a *separate hyperparameter that can smoothly average the observed iterates* at every iteration.

## 2.3 SCHEDULE-FREE LEARNING

In parallel, Schedule-Free learning (SF) (Defazio et al., 2024) was recently proposed as a wrapper to any base optimizer using a variant of the primal averaging formulation of Nesterov's method (equation 4) for hyperparameter-free learning:

$$
\begin{aligned}
y^{(t)} &= \mu x^{(t)} + (1 - \mu) z^{(t)} \\
z^{(t+1)} &= z^{(t)} - \gamma g(y^{(t)}; \xi^{(t)}) \\
x^{(t+1)} &= \frac{t}{t+1} x^{(t)} + \left(1 - \frac{t}{t+1}\right) z^{(t+1)}.
\end{aligned}
\tag{7}
$$

Originally designed to eliminate the need for manually specified learning rate schedules, Schedule-Free has demonstrated the surprising ability to not only match, but even surpass the practical performance of the original base optimizer. This is done by *decoupling* the momentum hyperparameter used in the $x^{(t)}$ and $y^{(t)}$ sequences, unlike the standard primal averaging formulation of Nesterov (equation 4). Through the choice of $\mu$, the method interpolates between uniform Polyak-Ruppert averaging and stochastic primal averaging (Ruppert, 1988; Polyak, 1990; Tao et al., 2018).

Ignoring the hyperparameter-free learning problem, one could alternatively replace uniform averaging with exponential moving averaging of the iterates, which is commonly used in practice (Morales-Brotons et al., 2024). This alternative suggests a different generalization of Nesterov momentum that may offer the potential flexibility necessary to reproduce DiLoCo's convergence gains without the two-level structure.

## 3 GENERALIZED PRIMAL AVERAGING (GPA)

By decoupling the constants for the model evaluation and gradient computation sequences in Nesterov's primal averaging formulation (equation 4) and leveraging the observation of using exponential moving averaging in place of uniform averaging in Schedule-Free (equation 7), we introduce the *Generalized Primal Averaging* (GPA) framework:

$$
\begin{aligned}
y^{(t)} &= \mu_y x^{(t)} + (1 - \mu_y) z^{(t)} \\
z^{(t+1)} &= z^{(t)} - \gamma^{(t)} g(y^{(t)}; \xi^{(t)}) \\
x^{(t+1)} &= \mu_x x^{(t)} + (1 - \mu_x) z^{(t+1)}.
\end{aligned}
\tag{8}
$$

Here, $\mu_x \in [0, 1)$ and $\mu_y \in [0, 1]$ are independent hyperparameters that separately control the degree of interpolation used to maintain the model evaluation sequence $x^{(t)}$ and gradient computation sequence $y^{(t)}$. The additional hyperparameter $\mu_x$ serves as a smoothening or exponential moving average parameter that replaces Polyak-Ruppert averaging in Schedule-Free, while $\mu_y$ controls the amount of information flow into $y^{(t)}$. The complete pseudocode for a general base optimizer is provided in Algorithm 1.

Unlike the modern formulation of Nesterov momentum (equation 3) or DiLoCo (equation 6) built on (pseudo-)gradient averaging, GPA is defined based on the *primal or iterate averaging framework*. We argue that this provides a more meaningful characterization of the method. For example, the primal averaging interpretation naturally extends to other search directions by replacing $-g(y^{(t)}; \xi^{(t)})$ with the search direction $d^{(t)}$ evaluated at $y^{(t)}$. This extension is not intuitive from the gradient averaging perspective, as it would translate to averaging search directions (with potentially different, evolving preconditioners) in the momentum buffer.

**Learning rate schedules.** By replacing Polyak-Ruppert averaging with exponential moving averaging, GPA is not inherently schedule-free and requires the use of a learning rate schedule. To see why, observe that Polyak averaging places increasingly less weight $1/(t + 1)$ on the most recent iterate $z^{(t+1)}$, which plays a similar role to learning rate scheduling (Sandler et al., 2023; Defazio et al., 2024). GPA instead places a constant weight $\mu_x$ on the most recent iterate $z^{(t+1)}$ by leveraging an exponential moving average. This is reflected theoretically in their last-iterate convergence properties.

---

**Algorithm 1** Generalized Primal Averaging (GPA)

---

1: **Input:** Initial iterate $x^{(1)}$, learning rate schedule $\gamma^{(t)} > 0$, weight decay $\lambda \geq 0$, interpolation parameters $\mu_x, \mu_y \in [0, 1)$, base optimizer `BaseOpt`.
2: $z^{(1)} = x^{(1)}$
3: **for** $t = 1, ..., T$ **do**
4: $\quad y^{(t)} = \mu_y x^{(t)} + (1 - \mu_y)z^{(t)}$ $\qquad\qquad$ ▷ Update gradient computation point $y^{(t)}$.
5: $\quad g^{(t)} \in \partial f(y^{(t)}; \xi^{(t)})$ $\qquad\qquad\qquad\qquad$ ▷ Gradient is evaluated at $y^{(t)}$.
6: $\quad d^{(t)} = \texttt{BaseOpt}(g^{(t)})$ $\qquad\qquad\qquad$ ▷ Compute base optimizer's search direction.
7: $\quad z^{(t+1)} = (1 - \gamma^{(t)}\lambda)z^{(t)} + \gamma^{(t)}d^{(t)}$ $\qquad\qquad\qquad$ ▷ Update $z^{(t)}$ iterate.
8: $\quad x^{(t+1)} = \mu_x x^{(t)} + (1 - \mu_x)z^{(t+1)}$ $\qquad\qquad$ ▷ Update weighted iterate average $x^{(t)}$.
9: **end for**
10: Return $x^{(T)}$

---

**Degenerate cases.** The choice of $\mu_x$ and $\mu_y$ enables GPA to recover different averaging methods. When $\mu_y = 1$, $x^{(t)} = y^{(t)}$ and we recover stochastic primal averaging, or equivalently, LaProp (Defazio, 2020; Ziyin et al., 2020); see Appendix C. When $\mu_y = 0$, $x^{(t)}$ and $z^{(t)} = y^{(t)}$ become decoupled and we recover exponential moving averaging of the iterates (Morales-Brotons et al., 2024). When $\mu_x = 0$, $x^{(t)} = y^{(t)} = z^{(t)}$ for any choice of $\mu_y$, and GPA reverts to the base optimizer.

**Other properties.** GPA also retains several desirable properties of the base optimizer for deep learning. Because $\mu_x, \mu_y \in [0, 1]$, GPA preserves modular norm bounds of the model parameters. Additionally, GPA requires only one extra copy of the model weights for implementation – specifically, by storing $y^{(t)}$ and reconstructing $x^{(t)}$ from $y^{(t)}$ and $z^{(t)}$ during evaluation – unlike DiLoCo, which demands more memory overhead. More details on these properties are provided in Appendix C.

## 3.1 INTERPRETING GPA AS SMOOTHENED DiLoCo

As seen in Figure 1a, increasing the number of inner steps leads to improved performance for DiLoCo in the non-distributed setup. However, the underlying reasons for this behavior are not understood. By examining DiLoCo from the lens of GPA in equation 8 and comparing it with the more restrictive Nesterov formulation in equation 4, we can develop a deeper intuition for DiLoCo's inner workings.

Suppose that we increase the number of inner steps in DiLoCo and want to maintain the same level of smoothing on the average iterate $x^{(t)}$. One may attempt to increase $\mu$ in Nesterov (equation 4) to decrease the weight on the current iterate $z^{(t+1)}$. However, since $\mu$ controls both the amount of smoothing in $x^{(t)}$ *and* the amount of interpolation used to update $y^{(t)}$, strictly increasing $\mu$ would *decrease the recency of information from $z^{(t)}$ in $y^{(t)}$* by a factor of $\mu^2$, resulting in significantly different algorithmic behavior. Numerically, we validate that tuning $\mu$ alone in Nesterov's primal averaging formulation is not sufficient to reach the performance of DiLoCo; see Appendix E.

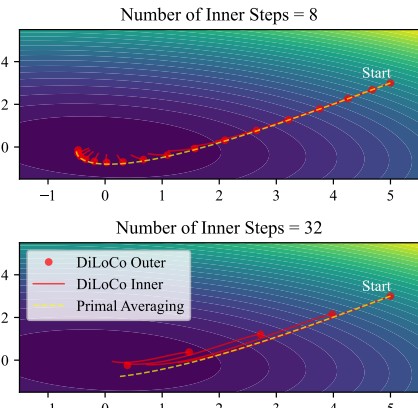

Figure 2: Comparison of DiLoCo and GPA's trajectories on a deterministic quadratic problem. The outer iterates of DiLoCo are shown as red points, and the inner iterates as thin red lines.

GPA addresses this limitation by decoupling the two roles of $\mu$ into separate hyperparameters: $\mu_x$ for the model evaluation sequence and $\mu_y$ for the gradient computation sequence. By controlling these two interpolation constants independently, we can smooth $x^{(t)}$ similarly without changing the amount of information introduced into $y^{(t)}$. This smoothing is depicted in Figure 2 on a simple

deterministic quadratic problem. For a small number of inner steps, the methods closely align, but for a larger number of inner steps, their behavior diverges.

**Tuning GPA from DiLoCo.** This intuition provides practical guidelines for converting a tuning for DiLoCo to GPA. Given an optimal number of inner steps $H$ and momentum parameter $\mu$ in DiLoCo, we observe for GPA that $x^{(t+H)} = \mu_x^H x^{(t)} + (1 - \mu_x) \sum_{k=0}^{H-1} \mu_x^k z^{(t+H-k)}$. Therefore, to match the coefficient in front of $x^{(t)}$ with DiLoCo, one can set $\mu_x = \mu^{1/H}$ while keeping $\mu_y \approx \mu$. With commonly used values $\mu = 0.9$ and $H = 32$, we obtain $\mu_x \approx 0.9967$ and $\mu_y \approx 0.9$. We leverage this heuristic to determine an effective number of inner steps used in Figure 1.

**Tradeoffs with DiLoCo.** GPA not only outperforms DiLoCo, but does so with fewer hyperparameters and lower memory requirements. While DiLoCo requires four hyperparameters, e.g., the inner and outer learning rate, momentum hyperparameter, and number of inner steps, GPA reduces this to just three: the learning rate and two momentum parameters. This simplification is possible because DiLoCo's practical performance is governed by an effective learning rate that couples the effect of the inner and outer learning rates ($\gamma^{(t)}$ and $\tilde{\gamma}$). On the other hand, GPA requires more FLOPs per-iteration, while DiLoCo amortizes its additional compute cost across multiple inner steps.

## 4 EXPERIMENTS

In this section, we assess the effectiveness of GPA on both language model pre-training and computer vision workloads. For language modeling, we compare against baselines AdamW and DiLoCo, while for computer vision experiments we compare GPA against AdamW. For both DiLoCo and GPA, we use AdamW as the base optimizer (DiLoCo-AdamW and GPA-AdamW, respectively).

### 4.1 LANGUAGE MODEL PRE-TRAINING

We conduct experiments on two scales of Llama models: (1) **160 million parameters** and (2) **1 billion parameters**. These are pre-trained on the C4 dataset from scratch (Raffel et al., 2019) using a token budget of roughly 3.2 billion and 50 billion tokens, respectively (Hoffmann et al., 2022). All of our small experiments are conducted on a single machine equipped with eight H100 GPUs (97 GB of memory) while the large scale model experiments utilize two nodes (with a total of 16 GPUs). Comprehensive details on batch size, sequence length, and hyperparameter sweeps can be found in Appendix E. Note that the Llama-1B experiments are performed in an overtrained setting.

Table 1: Final validation loss versus effective number of inner steps $H$ for different optimizers on **Llama-160M** and **Llama-1B** models.

| | Llama-160M | | | Llama-1B | | | |
|---|---|---|---|---|---|---|---|
| Method | $H = 8$ | $H = 16$ | $H = 32$ | $H = 16$ | $H = 32$ | $H = 64$ | $H = 128$ |
| AdamW | 3.3561 | 3.3561 | 3.3561 | 2.6886 | 2.6886 | 2.6886 | 2.6886 |
| DiLoCo-AdamW | 3.2977 | 3.2804 | 3.3037 | 2.6835 | 2.6765 | 2.6755 | 2.6743 |
| GPA-AdamW | 3.2769 | **3.2595** | 3.2774 | 2.6828 | 2.6722 | **2.6619** | 2.6734 |

**Performance across number of inner steps.** In Table 1, we provide the final validation loss for each method for different effective number of inner steps. Consistent with Figure 1a, GPA-AdamW supersedes both DiLoCo and AdamW, except when the number of inner steps is 1. Both DiLoCo and GPA display U-shaped behavior with respect to the number of inner steps, and share a similar optimal effective number of inner steps, validating our heuristic on the choice of $\mu_x$.

**Convergence behavior.** Figure 3 shows the validation loss curves on Llama-160M for AdamW, DiLoCo-AdamW, and GPA-AdamW for the case where the number of inner steps is 16. In this case, $\mu_x$ has been tuned to match the number of inner steps; see Table 3 in Appendix E for details. GPA-AdamW converges faster than both DiLoCo and AdamW throughout the entire training run. The training curves for GPA-AdamW are also noticeably smoother and more stable compared to the other methods. Our hyperparameter sweeps reveal that GPA-AdamW can handle higher learning rates compared to DiLoCo and AdamW, e.g., $5 \cdot 10^{-3}$.

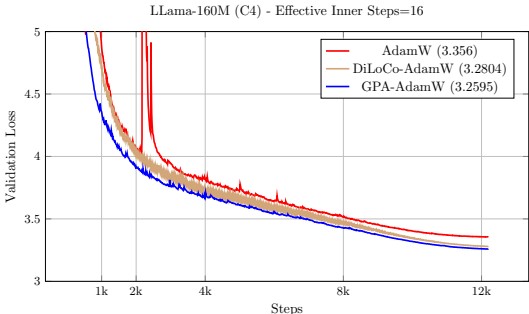

Figure 3: Validation loss vs steps for AdamW, DiLoCo, and GPA on Llama-160M.

## 4.2 VISION TRANSFORMER MODEL TRAINING

To validate our method on a computer vision task, we train a ViT-S/16 model from `timm` on ImageNet with data augmentations from the repository (see Figure 4). We use 8 random seeds for the runs. Our evaluation in both small batch (4,196) and large batch (16,384) settings indicate that GPA outperforms AdamW by a clear margin throughout the course of training. For further details on the hyperparameters used and performance in the large batch setting, see Appendix E.

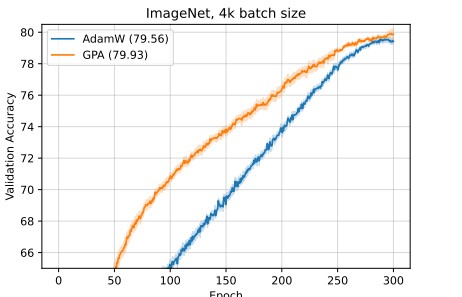
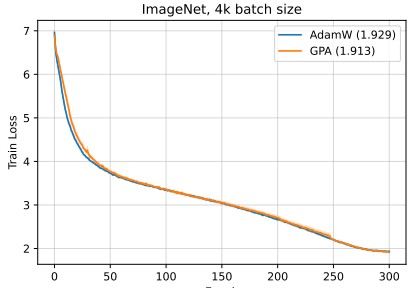

Figure 4: Comparison of AdamW and GPA on ImageNet ViT-S/16 from `timm` with data augmentations. The optimal configuration for both AdamW and GPA use a learning rate of 0.005 and weight decay of 0.1.

## 5 CONVERGENCE THEORY

Using the theoretical developments underpinning Schedule-Free learning, we can derive a convergence bound for GPA given any base optimizer that has a regret bound, using the framework of online-to-batch conversion (Cesa-Bianchi et al., 2004). We will use the Bregman divergence of $F$ defined as $B_F(a, b) = F(a) - F(b) - \langle \nabla F(b), a - b \rangle$ for $a, b \in \mathbb{R}^n$.

**Theorem 1.** *Let $F$ be a convex function and assume that there exists a minimizer $x_*$ that minimizes $F$. Let $\xi^{(1)}, \ldots, \xi^{(T)}$ be a sequence of i.i.d. random variables. Suppose that we are given arbitrary updates $z^{(1)}, \ldots, z^{(T)}$ from a base optimizer within the Generalized Primal Averaging framework*

*(Equation 8). Then for $\mu_x, \mu_y \in [0, 1)$ and average iterate $\bar{x}^{(T)} = \frac{1}{T} \sum_{t=1}^{T} x^{(t)}$, we have the bound*

$$\mathbb{E}[F(\bar{x}^{(T)}) - F(x_*)] \leq \frac{1}{T} \sum_{t=1}^{T} \mathbb{E}[\langle \nabla F(y^{(t)}), z^{(t)} - x_* \rangle] + \frac{\mu_x}{1 - \mu_x} \frac{1}{T} \mathbb{E}\left[ F(x^{(1)}) - F(x_*) \right]$$

$$- \frac{1}{1 - \mu_y} \frac{1}{T} \sum_{t=1}^{T} \mathbb{E}[B_F(y^{(t)}, x^{(t)})] - \frac{\mu_y}{1 - \mu_y} \frac{1}{T} \sum_{t=1}^{T} \mathbb{E}[B_F(x^{(t)}, y^{(t)})]$$

$$- \frac{\mu_x}{1 - \mu_x} \frac{1}{T} \sum_{t=1}^{T} \mathbb{E}[B_F(x^{(t-1)}, x^{(t)})].$$

**Corollary 1.** *Assume that the base optimizer has the regret guarantee $\sum_{t=1}^{T} \mathbb{E}[\langle \nabla F(y^{(t)}), z^{(t)} - x_* \rangle] = \mathcal{O}(\sqrt{T})$. Then:*

$$\mathbb{E}[F(\bar{x}^{(T)}) - F(x_*)] = \mathcal{O}\left( \frac{1}{\sqrt{T}} \right).$$

**Remarks on Theorem 1**:

- The first term on the right-hand side of the regret bound is the average regret of the base optimizer. This term captures the convergence rate from the base optimizer.

- The second term has a positive term, which decays at a rate of $1/T$, which is typically faster than the decay of the term in the first row.

- All remaining Bregman divergence terms are negative, and so are potentially beneficial. If $\mu_x$ and $\mu_y$ are chosen such that the negative terms dominate the positive second term, then GPA will converge faster than the base optimizer.

- The same terms appear in the convergence guarantees for Schedule-Free methods, and can explain when they may work better. For strongly convex problems, such Bregman divergences were used to get $\mathcal{O}(1/T)$ convergence.

- Unlike the guarantees for Schedule-Free, our convergence bound is for the average iterate. For best performance, a learning rate schedule should be used and the last iterate returned (Defazio et al., 2023).

- Our bound indicates that GPA will be faster than the base optimizer when the objective function varies nonlinearly between consecutive iterates and between $x^{(t)}$ and $y^{(t)}$.

## 6 CONCLUSION

GPA introduces independent interpolation constants for the gradient computation and model evaluation sequences that yield a flexible generalization of Nesterov momentum. On small-scale dense models, this flexibility allows GPA to outperform DiLoCo, while removing the complexity of its two-loop structure, simplifying hyperparameter tuning and reducing memory requirements in non-distributed settings.

Future work should validate GPA at scale across diverse model architectures and modalities and explore its compatibility with other base optimizers (e.g., Shampoo, SOAP, Muon) and hyperparameter transfer techniques such as $\mu$P (Yang & Hu, 2021; Yang et al., 2022). Additionally, while our convergence bound partially explains the empirical results, it is limited to the convex setting and does not fully characterize when GPA can outperform the base optimizer.

Finally, GPA's decoupling of parameters also enables new avenues for distributed training. In DiLoCo, the number of inner steps serves as a coupled hyperparameter for both Lookahead with Nesterov and local SGD, leading to the undesirable finding that increasing the number of inner steps can improve convergence – contrary to standard local SGD intuition. GPA introduces a tunable, continuous smoothing parameter that is independent of the number of local SGD steps, laying a new foundation for re-designing DiLoCo for cross-regional training.

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

# A    LLM Usage

We used an internal AI assistant for revising the grammar and wording in the paper, and used Gemini Pro 2.5 to verify our proofs.

# B    Formulations of Polyak Momentum

Similar to Nesterov momentum, classical or Polyak momentum also have different formulations that are commonly used in the community. The most commonly implemented formulation (which we call the *modern formulation*) is given as:

$$
\begin{aligned}
b^{(t)} &= \mu b^{(t-1)} + g(x^{(t)}; \xi^{(t)}), \\
x^{(t+1)} &= x^{(t)} - \gamma^{(t)} b^{(t)}.
\end{aligned}
\tag{9}
$$

The method accumulates a momentum buffer similar to Nesterov's modern formulation (equation 3), but only updates the weights using $b^{(t)}$ as opposed to $\mu b^{(t)} + g(x^{(t)}; \xi^{(t)})$.

This formulation can be re-written in the *heavy ball formulation*

$$
x^{(t+1)} = x^{(t)} - \gamma^{(t)} b^{(t)} + \mu(x^{(t)} - x^{(t-1)}),
\tag{10}
$$

which is also equivalent to the *primal averaging formulation* (Defazio, 2020)

$$
\begin{aligned}
z^{(t+1)} &= z^{(t)} - \gamma^{(t)} g(x^{(t)}; \xi^{(t)}), \\
x^{(t+1)} &= \mu x^{(t)} + (1 - \mu) z^{(t+1)}.
\end{aligned}
\tag{11}
$$

**Remarks.**

- The LaProp algorithm (Ziyin et al., 2020) uses the heavy ball formulation to motivate the generalization of momentum to preconditioned gradient methods by replacing the gradient $g(x^{(t)}; \xi^{(t)})$ with the search direction $d^{(t)}$ in equation 9.

- The primal averaging formulations for Polyak momentum (equation 11) and Nesterov momentum (equation 4) differ in their inclusion of the $y^{(t)}$ interpolated sequence, which determines where the gradient is evaluated. This is also reflected in Sutskever's formulation (equation 2).

- Polyak momentum can therefore be recovered by setting $\mu_y = 0$ in GPA (equation 8).

# C    Algorithmic Details

## C.1    Pseudocode for Non-Distributed DiLoCo / Lookahead with Nesterov

We provide a complete description of non-distributed DiLoCo in Algorithm 2.

---

**Algorithm 2** Non-Distributed DiLoCo / Lookahead with Nesterov

---

1: **Input:** Initial iterate $x^{(1)}$, inner learning rate schedule $\gamma^{(t)} > 0$, constant outer learning rate $\tilde{\gamma} > 0$, weight decay $\lambda \geq 0$, momentum parameter $\mu \in [0, 1)$, base optimizer `BaseOpt`.
2: $\tilde{x}^{(1)} = x^{(1)}$         ▷ Initialize slow model weights.
3: $b^{(0)} = 0 \in \mathbb{R}^n$         ▷ Initialize momentum buffer.
4: **for** step $t = 1, ..., T$ **do**
5:     Sample mini-batch $\xi^{(t)}$
6:     $g^{(t)} \in \partial f(x^{(t)}; \xi^{(t)})$
7:     $d^{(t)} = \texttt{BaseOpt}(g^{(t)})$         ▷ Computes base optimizer's search direction.
8:     $x^{(t+1)} = (1 - \gamma^{(t)}\lambda)x^{(t)} + \gamma^{(t)}d^{(t)}$    ▷ Updates inner model weights (with weight decay).
9:     **if** $t \bmod H = 0$ **then**
10:       $p^{(t)} = \tilde{x}^{(t)} - x^{(t+1)}$         ▷ Pseudo-gradient computation.
11:       $b^{(t+1)} = \mu b^{(t)} + p^{(t)}$         ▷ Accumulates outer momentum.
12:       $\tilde{x}^{(t+1)} = \tilde{x}^{(t)} - \tilde{\gamma}\left[\mu b^{(t)} + p^{(t)}\right]$      ▷ Nesterov-style parameter update.
13:       $x^{(t+1)} = \tilde{x}^{(t+1)}$         ▷ Re-initialize inner model weights.
14:     **else**
15:       $\tilde{x}^{(t+1)} = \tilde{x}^{(t)}$
16:       $b^{(t+1)} = b^{(t)}$
17:     **end if**
18: **end for**
19: **Returns:** $\tilde{x}^{(T)}$

---

## C.2 MEMORY-EFFICIENT FORMULATION OF GENERALIZED PRIMAL AVERAGING

The implementation of the original formulation of GPA in equation 8 requires storing two additional copies of the model's parameters during the optimizer step. This is because the gradient computation occurs on the $y^{(t)}$ sequence, which is computed from the two other sequences $x^{(t)}$ and $z^{(t)}$. To avoid this additional model copy, we can store $y^{(t)}$ instead, and recover $x^{(t)}$ from $y^{(t)}$ and $z^{(t)}$ during evaluation time.

To see how this can be done, we define the *memory-efficient formulation* of GPA as:

$$
\begin{aligned}
x^{(t)} &= \frac{1}{\mu_y}y^{(t)} + \left(1 - \frac{1}{\mu_y}\right)z^{(t)}, \\
y^{(t+1)} &= \mu_x y^{(t)} + (1 - \mu_x)z^{(t)} - (1 - \mu_x\mu_y)\gamma^{(t)}g(y^{(t)}; \xi^{(t)}), \\
z^{(t+1)} &= z^{(t)} - \gamma^{(t)}g(y^{(t)}; \xi^{(t)}).
\end{aligned}
\tag{12}
$$

This reformulation is valid only when $\mu_y > 0$. In the $y^{(t)}$ update, the first term can be interpreted as interpolating $y^{(t)}$ towards $z^{(t)}$. The second term is a correction term that applies a dampened update on $y^{(t)}$.

Note that this formulation does not require the computation of $x^{(t)}$ except when necessary. Therefore, our implementation enables a training and evaluation mode similar to neural network modules like batch normalization that enables us to compute $x^{(t)}$ from $y^{(t)}$ and vice-versa. Specifically, when switching from training to evaluation mode, we can compute $x^{(t)}$ from $y^{(t)}$ and $z^{(t)}$ by:

$$
x^{(t)} = \frac{1}{\mu_y}y^{(t)} + \left(1 - \frac{1}{\mu_y}\right)z^{(t)}.
$$

Similarly, when switching from evaluation to training mode, we can recover $y^{(t)}$ from $x^{(t)}$ and $z^{(t)}$ by:

$$
y^{(t)} = \mu_y x^{(t)} + (1 - \mu_y)z^{(t)}.
$$

A proof of the equivalence of these two formulations is provided in Appendix D. The complete pseudocode for arbitrary base optimizers are provided in Algorithm 3.

---

**Algorithm 3** Memory-Efficient Generalized Primal Averaging (GPA)

---

1: **Input:** Initial iterate $y^{(1)}$, learning rate schedule $\gamma^{(t)} > 0$, weight decay $\lambda \geq 0$, interpolation parameters $\mu_x, \mu_y \in [0, 1)$, base optimizer `BaseOpt`.
2: $z^{(1)} = y^{(1)}$
3: **for** $t = 1, ..., T$ **do**
4: $\quad g^{(t)} \in \partial f(y^{(t)}; \xi^{(t)})$
5: $\quad d^{(t)} = \texttt{BaseOpt}(g^{(t)})$
6: $\quad y^{(t)} = \mu_x y^{(t)} + (1 - \mu_x) z^{(t)} + \gamma^{(t)}(1 - \mu_x \mu_y)(d^{(t)} + \lambda z^{(t)})$
7: $\quad z^{(t+1)} = (1 - \gamma^{(t)} \lambda) z^{(t)} - \gamma^{(t)} d^{(t)}$
8: **end for**
9: **Returns:** $x^{(T)} = \frac{1}{\mu_y} y^{(T)} + \left(1 - \frac{1}{\mu_y}\right) z^{(T)}$.

---

### C.3 COMPATIBILITY WITH MODULAR NORM THEORY

Recent work on Muon and similar methods have built on modular norm theory, which suggests that the design of optimization methods for deep learning should constrain the modular norm of the model parameters in order to enable hyperparameter transferability and bounded Lipschitz constants (Large et al., 2024; Jordan et al., 2024; Pethick et al., 2025). Here, we argue that GPA, by definition, preserves these norm constraints.

To see this, assume that $d^{(t)}$ is the search direction for a single parameter that it is constrained with respect to some norm, i.e., $\|d^{(t)}\| \leq M$ for some constant $M \geq 0$. (Typically, we assume it is the RMS-to-RMS norm or similar.) We can preserve these norm constraints on the iterates produced by GPA since:

$$\|y^{(t)}\| \leq \mu_y \|x^{(t)}\| + (1 - \mu_y) \|z^{(t)}\|$$
$$\|z^{(t+1)}\| \leq (1 - \lambda \gamma^{(t)}) \|z^{(t)}\| + \gamma^{(t)} \|d^{(t)}\|$$
$$\|x^{(t+1)}\| \leq \mu_x \|x^{(t)}\| + (1 - \mu_x) \|z^{(t+1)}\|.$$

Since $\mu_x, \mu_y \in [0, 1]$, we can see that if $\max\left\{\|x^{(t)}\|, \|y^{(t)}\|, \|z^{(t)}\|\right\} \leq M'$ for $M' \geq 0$, then $\max\left\{\|x^{(t+1)}\|, \|y^{(t+1)}\|, \|z^{(t+1)}\|\right\} \leq (1 - \lambda \gamma^{(t)}) M' + \gamma^{(t)} M$, which is the same bound we would obtain for the base optimizer.

## D PROOFS

### D.1 EQUIVALENCE BETWEEN NESTEROV'S FORMULATIONS

**Proposition 2.** *Given fixed learning rates $\gamma_{\text{primal}}, \gamma_{\text{modern}} > 0$, Nesterov's primal averaging formulation (equation 4) is equivalent to Nesterov's modern formulation (equation 3) in the sense that*

$$y_{\text{primal}}^{(t)} = x_{\text{modern}}^{(t)} \quad and \quad b_{\text{modern}}^{(t)} = \frac{1}{(1 - \mu) \gamma_{\text{primal}}} \left(x_{\text{primal}}^{(t)} - x_{\text{primal}}^{(t+1)}\right), \tag{13}$$

*when $\mu_{\text{primal}} = \mu_{\text{modern}} = \mu$ and $(1 - \mu) \gamma_{\text{primal}} = \gamma_{\text{modern}}$.*

*Proof.* We can prove this by induction. For simplicity of notation, we will use $x_m = x_{\text{modern}}$ and $x_p = x_{\text{primal}}$ and similar for all variables.

For the base case, note that the initializations $z_p^{(1)} = x_p^{(1)} = x_m^{(1)}$ are equal. Therefore,

$$y_p^{(1)} = \mu x_p^{(1)} + (1 - \mu) z_p^{(1)} = x_m^{(1)}, \tag{14}$$

as desired. In addition, since $b_m^{(1)} = \mu b_m^{(0)} + g(x_m^{(1)}; \xi^{(1)}) = g(x_m^{(1)})$, we can see that:

$$
\begin{aligned}
x_p^{(1)} - x_p^{(2)} &= (1 - \mu)x_p^{(1)} - (1 - \mu)z_p^{(1)} \\
&= (1 - \mu)(x_p^{(1)} - z_p^{(2)}) \\
&= (1 - \mu)(x_p^{(1)} - z_p^{(1)} + \gamma_p g(y_p^{(1)}; \xi^{(1)})) \\
&= (1 - \mu)\gamma_p g(y_p^{(1)}; \xi^{(1)}).
\end{aligned}
$$

The base case for the momentum buffer $b_m^{(1)}$ follows from rearranging the equation with equation 14 and observing that $b_m^{(1)} = \mu b_m^{(0)} + g(x_m^{(1)}; \xi^{(1)}) = g(x_m^{(1)}; \xi^{(1)})$.

For the inductive step, assume that equation 13 holds for $t$. Then from the inductive hypothesis, we can show that:

$$
\begin{aligned}
x_m^{(t+1)} &= x_m^{(t)} - \gamma_m[\mu b_m^{(t)} + g(x_m^{(t)}; \xi^{(t)})] \\
&= y_p^{(t)} - (1 - \mu)\gamma_p \left[ \mu \left( \frac{1}{(1 - \mu)\gamma_p}(x_p^{(t)} - x_p^{(t+1)}) \right) + g(y_p^{(t)}; \xi^{(t)}) \right] \\
&= y_p^{(t)} - \mu(x_p^{(t)} - x_p^{(t+1)}) - (1 - \mu)\gamma g(y_p^{(t)}; \xi^{(t)}).
\end{aligned}
\tag{15}
$$

From the primal averaging form in equation 4, we can derive that:

$$
\begin{aligned}
x_p^{(t+1)} &= \mu x_p^{(t)} + (1 - \mu)z_p^{(t+1)} \\
&= \mu x_p^{(t)} + (1 - \mu)(z_p^{(t)} - \gamma_p g(y_p^{(t)}; \xi^{(t)}) \\
&= y_p^{(t)} - (1 - \mu)\gamma_p g(y_p^{(t)}; \xi^{(t)}).
\end{aligned}
\tag{16}
$$

Rearranging equation 16, we get that:

$$
y_p^{(t)} - x_p^{(t+1)} = (1 - \mu)\gamma_p g(y_p^{(t)}; \xi^{(t)}).
\tag{17}
$$

Plugging in equation 17 into equation 15, we obtain:

$$
x_m^{(t+1)} = y_p^{(t)} - \mu(x_p^{(t)} - x_p^{(t+1)}) - (y_p^{(t)} - x_p^{(t+1)}) = (1 + \mu)x_p^{(t+1)} - \mu x_p^{(t)}.
\tag{18}
$$

Finally, since $x_p^{(t+1)} = \mu x_p^{(t)} + (1 - \mu)z_p^{(t)}$, $(1 - \mu)z_p^{(t+1)} = x_p^{(t+1)} - \mu x_p^{(t)}$. Therefore, to see $x_m^{(t+1)}$'s equivalence to $y_p^{(t+1)}$,

$$
\begin{aligned}
y_p^{(t+1)} &= \mu x_p^{(t+1)} + (1 - \mu)z_p^{(t+1)} \\
&= \mu x_p^{(t+1)} + x_p^{(t+1)} - \mu x_p^{(t)} \\
&= (1 + \mu)x_p^{(t+1)} - \mu x_p^{(t)}.
\end{aligned}
\tag{19}
$$

Combining equations 18 and 19 gives the result.

To prove that $b_m^{(t+1)} = \frac{1}{(1-\mu)\gamma_p}(x_p^{(t+1)} - x_p^{(t+2)})$, note that:

$$
b_m^{(t+1)} = \mu b_m^{(t)} + g(x_m^{(t+1)}; \xi^{(t+1)}) = \frac{\mu}{(1 - \mu)\gamma_p}(x_p^{(t)} - x_p^{(t+1)}) + g(y_p^{(t+1)}; \xi^{(t+1)}).
\tag{20}
$$

To get an expression for $x_p^{(t+1)} - x_p^{(t+2)}$, note that:

$$
\begin{aligned}
x_p^{(t+2)} &= \mu x_p^{(t+1)} + (1 - \mu)(z_p^{(t+1)} - \gamma_p g(y_p^{(t+1)}; \xi^{(t+1)})) \\
&= (\mu x_p^{(t+1)} + (1 - \mu)z_p^{(t+1)}) - (1 - \mu)\gamma_p g(y_p^{(t+1)}; \xi^{(t+1)}) \\
&= y_p^{(t+1)} - (1 - \mu)\gamma_p g(y_p^{(t+1)}; \xi^{(t+1)}) \\
&= ((1 + \mu)x_p^{(t+1)} - \mu x_p^{(t)}) - (1 - \mu)\gamma_p g(y_p^{(t+1)}; \xi^{(t+1)}),
\end{aligned}
\tag{21}
$$

where equation 21 follows from equation 19. Therefore, plugging-in equation 21 into $x_p^{(t+1)} - x_p^{(t+2)}$ gives:

$$
x_p^{(t+1)} - x_p^{(t+2)} = -\mu(x_p^{(t+1)} - x_p^{(t)}) + (1 - \mu)\gamma_p g(y_p^{(t+1)}; \xi^{(t+1)}).
\tag{22}
$$

The result follows from expanding equation 20 as:

$$b_m^{(t+1)} = \frac{1}{(1-\mu)\gamma_p} \left[ -\mu(x_p^{(t+1)} - x_p^{(t)}) + (1-\mu)\gamma_p g(y_p^{(t+1)}; \xi^{(t+1)}) \right]$$

$$= \frac{1}{(1-\mu)\gamma_p}(x_p^{(t+1)} - x_p^{(t+2)}).$$

$\square$

## D.2 EQUIVALENCE BETWEEN GENERALIZED PRIMAL AVERAGING FORMULATIONS

**Proposition 3.** *Let $\mu_y > 0$. Then GPA (equation 8) is equivalent to the memory-efficient formulation (equation 12).*

*Proof.* Note that it is sufficient to show that:

$$x^{(t)} = \frac{1}{\mu_y}y^{(t)} + \left(1 - \frac{1}{\mu_y}\right)z^{(t)}, \tag{23}$$

$$y^{(t+1)} = \mu_x y^{(t)} + (1-\mu_x)z^{(t)} - (1-\mu_x\mu_y)\gamma^{(t)}g(y^{(t)}; \xi^{(t)}). \tag{24}$$

To prove equation 23, note that we can re-write $x^{(t)}$ as a function of $y^{(t)}$ and $z^{(t)}$, i.e., since

$$y^{(t)} = \mu_y x^{(t)} + (1-\mu_y)z^{(t)}$$

and $\mu_y > 0$, we have that

$$x^{(t)} = \frac{1}{\mu_y}y^{(t)} + \frac{1}{\mu_y}(\mu_y - 1)z^{(t)} = \frac{1}{\mu_y}y^{(t)} + \left(1 - \frac{1}{\mu_y}\right)z^{(t)}.$$

To prove equation 23, we can re-write equation 23 as

$$\mu_y x^{(t+1)} = \mu_y z^{(t+1)} + (y^{(t+1)} - z^{(t+1)}) = y^{(t+1)} - (1-\mu_y)z^{(t+1)}. \tag{25}$$

Similarly, by plugging in the original $x^{(t+1)}$ update, i.e., $x^{(t+1)} = \mu_x x^{(t)} + (1-\mu_x)z^{(t)}$, we also have:

$$\mu_y x^{(t+1)} = \mu_y(\mu_x x^{(t)} + (1-\mu_x)z^{(t)}) = \mu_x \mu_y x^{(t)} + (1-\mu_x)\mu_y z^{(t+1)}. \tag{26}$$

Combining these two equalities in equations 25 and 26 and rearranging, we get:

$$y^{(t+1)} = \mu_x \mu_y x^{(t)} + (1-\mu_x\mu_y)z^{(t+1)}. \tag{27}$$

Plugging-in equation 23 and the update $z^{(t+1)} = z^{(t)} - \gamma^{(t)}g(y^{(t)}; \xi^{(t)})$ from equation 8 into equation 27, we obtain:

$$y^{(t+1)} = \mu_x\mu_y\left(\frac{1}{\mu_y}y^{(t)} + \left(1 - \frac{1}{\mu_y}\right)z^{(t)}\right) + (1-\mu_x\mu_y)(z^{(t)} - \gamma^{(t)}g(y^{(t)}; \xi^{(t)}))$$

$$= \mu_x y^{(t)} + (1-\mu_x)z^{(t)} - (1-\mu_x\mu_y)\gamma^{(t)}g(y^{(t)}; \xi^{(t)}),$$

as desired. $\square$

## D.3 CONVERGENCE BOUNDS BASED ON ONLINE-TO-BATCH THEORY

Our proofs similarly rely on the online-to-batch conversion theory used in Defazio et al. (2024).

**Lemma 1.** *Suppose we define $w^{(t)}$ as the weighting:*

$$w^{(t)} = \begin{cases} 1 & \text{if } t = 1, \\ (1-\mu_x)\mu_x^{-t+1} & \text{if } t > 1. \end{cases}$$

*Then the model evaluation sequence $x^{(t)}$ is equivalent to the weighted average:*

$$x^{(t+1)} = \frac{\sum_{i=1}^{t} w^{(i)}}{\sum_{i=1}^{t+1} w^{(i)}}x^{(t)} + \frac{w^{(t+1)}}{\sum_{i=1}^{(t+1)} w^{(i)}}z^{(t+1)} = \frac{w^{(1:t)}}{w^{(1:t+1)}}x^{(t)} + \frac{w^{(t+1)}}{w^{(1:t+1)}}z^{(t+1)},$$

*with*

$$w^{(1:t)} = \sum_{s=1}^{t} w^{(s)} = \mu_x^{-t+1}.$$

*Furthermore, $x^{(t)}$ can be expressed as the closed form expression:*

$$x^{(t)} = \mu_x^{t-1} \sum_{s=1}^{t} w^{(s)} z^{(s)}.$$

**Theorem 2.** *Let $F$ be a convex function, and assume that there exists a minimizer $x_*$ that minimizes $F$. Let $\xi^{(1)}, \ldots, \xi^{(T)}$ be a sequence of i.i.d. random variables. Suppose that we are given arbitrary updates $z^{(1)}, \ldots, z^{(T)}$ from a base optimizer within the Generalized Primal Averaging framework (Equation 8). Then for $\mu_x, \mu_y \in [0, 1)$ and average iterate $\bar{x}^{(T)} = \frac{1}{T} \sum_{t=1}^{T} x^{(t)}$, we have the bound*

$$\mathbb{E}[F(\bar{x}^{(T)}) - F(x_*)] \le \frac{1}{T} \sum_{t=1}^{T} \mathbb{E}[\langle \nabla F(y^{(t)}), z^{(t)} - x_* \rangle]$$

$$+ \frac{\mu_x}{1 - \mu_x} \frac{1}{T} \mathbb{E}\left[ F(x^{(1)}) - F(x_*) \right]$$

$$- \frac{1}{1 - \mu_y} \frac{1}{T} \sum_{t=1}^{T} \mathbb{E}[B_F(y^{(t)}, x^{(t)})] - \frac{\mu_y}{1 - \mu_y} \frac{1}{T} \sum_{t=1}^{T} \mathbb{E}[B_F(x^{(t)}, y^{(t)})]$$

$$- \frac{\mu_x}{1 - \mu_x} \frac{1}{T} \sum_{t=1}^{T} \mathbb{E}[B_F(x^{(t-1)}, x^{(t)})].$$

*Proof.* We start with the same analysis as in the Schedule-Free work (Defazio et al., 2024). Notice that by definition of $x^{(t)}$, it holds $w^{(1:t-1)}(x^{(t)} - x^{(t-1)}) = w^{(t)}(z^{(t)} - x^{(t)})$. Therefore,

$$w^{(1:t)} F(x^{(t)}) - w^{(1:t-1)} F(x^{(t-1)}) - w^{(t)} F(x_*)$$
$$= w^{(1:t-1)}(F(x^{(t)}) - F(x^{(t-1)})) + w^{(t)}(F(x^{(t)}) - F(x_*))$$
$$= w^{(1:t-1)}(\langle \nabla F(x^{(t)}), x^{(t)} - x^{(t-1)} \rangle - B_F(x^{(t-1)}, x^{(t)})) + w^{(t)}(F(x^{(t)}) - F(x_*))$$
$$= w^{(t)} \langle \nabla F(x^{(t)}), z^{(t)} - x^{(t)} \rangle - w^{(1:t-1)} B_F(x^{(t-1)}, x^{(t)}) + w^{(t)}(F(x^{(t)}) - F(x_*)).$$

Next, we observe that by definition of $y^{(t)}$, it holds $z^{(t)} - y^{(t)} = \frac{\mu_y}{1 - \mu_y}(y^{(t)} - x^{(t)})$, and, thus,

$$\langle \nabla F(x^{(t)}), z^{(t)} - x^{(t)} \rangle$$
$$= \langle \nabla F(x^{(t)}) - \nabla F(y^{(t)}), z^{(t)} - y^{(t)} \rangle + \langle \nabla F(y^{(t)}), z^{(t)} - y^{(t)} \rangle$$
$$\quad + \langle \nabla F(x^{(t)}), y^{(t)} - x^{(t)} \rangle$$
$$= \frac{\mu_y}{1 - \mu_y} \langle \nabla F(x^{(t)}) - \nabla F(y^{(t)}), y^{(t)} - x^{(t)} \rangle + F(x_*) - F(y^{(t)}) - B_F(x_*, y^{(t)}) + \langle \nabla F(y^{(t)}), z^{(t)} - x_* \rangle$$
$$\quad + F(y^{(t)}) - F(x^{(t)}) - B_F(y^{(t)}, x^{(t)})$$
$$\le -\frac{\mu_y}{1 - \mu_y} (B_F(x^{(t)}, y^{(t)}) + B_F(y^{(t)}, x^{(t)})) + F(x_*) - F(x^{(t)}) - B_F(y^{(t)}, x^{(t)}) + \langle \nabla F(y^{(t)}), z^{(t)} - x_* \rangle$$
$$= -\frac{\mu_y}{1 - \mu_y} B_F(x^{(t)}, y^{(t)}) - \frac{1}{1 - \mu_y} B_F(y^{(t)}, x^{(t)}) + F(x_*) - F(x^{(t)}) + \langle \nabla F(y^{(t)}), z^{(t)} - x_* \rangle,$$

where the inequality step used $-B_F(x_*, y^{(t)}) \le 0$, which follows from convexity of $F$. Plugging this back, we obtain

$$w^{(1:t)}F(x^{(t)}) - w^{(1:t-1)}F(x^{(t-1)}) - w^{(t)}F(x_*)$$

$$\le -w^{(t)}\frac{\mu_y}{1-\mu_y}B_F(x^{(t)}, y^{(t)}) - \frac{w^{(t)}}{1-\mu_y}B_F(y^{(t)}, x^{(t)}) + w^{(t)}(F(x_*) - F(x^{(t)}))$$

$$+ w^{(t)}\langle\nabla F(y^{(t)}), z^{(t)} - x_*\rangle - w^{(1:t-1)}B_F(x^{(t-1)}, x^{(t)}) + w^{(t)}(F(x^{(t)}) - F(x_*))$$

$$= w^{(t)}\langle\nabla F(y^{(t)}), z^{(t)} - x_*\rangle - \frac{w^{(t)}}{1-\mu_y}B_F(y^{(t)}, x^{(t)})$$

$$- \frac{w^{(t)}\mu_y}{1-\mu_y}B_F(x^{(t)}, y^{(t)}) - w^{(1:t-1)}B_F(x^{(t-1)}, x^{(t)}). \tag{28}$$

We may adapt this bound to our setting by using an exponentially increasing weighting sequence, given by Lemma 1. Using those weights, we have simplified expressions for the following quantities:

$$\frac{w^{(1:t)}}{w^{(t)}} = \frac{\mu_x^{-t+1}}{(1-\mu_x)\mu_x^{-t+1}} = \frac{1}{1-\mu_x},$$

$$\frac{w^{(1:t-1)}}{w^{(t)}} = \frac{\mu_x^{-(t-1)+1}}{(1-\mu_x)\mu_x^{-t+1}} = \frac{\mu_x}{1-\mu_x},$$

with a special case for the first iterate $\frac{w^{(1:1)}}{w^{(1)}} = 1$ and $\frac{w^{(1:t-1)}}{w^{(1)}} = 0$.

To obtain an average regret bound, we divide Equation 28 by $w^{(t)}$, take expectation, and sum from 1 to $T$. The left-hand side is a telescoping sum, which we can simplify as follows:

$$\sum_{t=1}^{T}\left[\frac{w^{(1:t)}}{w^{(t)}}\mathbb{E}[F(x^{(t)})] - \frac{w^{(1:t-1)}}{w^{(t)}}\mathbb{E}[F(x^{(t-1)})]\right] - TF(x_*)$$

$$= F(x^{(1)}) - \frac{w^{(1:1)}}{w^{(2)}}F(x^{(1)}) + \frac{1}{1-\mu_x}\sum_{t=2}^{T}\mathbb{E}[F(x^{(t)})] - \frac{\mu_x}{1-\mu_x}\sum_{t=2}^{T-1}\mathbb{E}[F(x^{(t)})] - TF(x_*)$$

$$= F(x^{(1)}) - \frac{1}{(1-\mu_x)\mu_x^{-1}}F(x^{(1)}) + \frac{1}{1-\mu_x}\mathbb{E}[F(x^{(T)})] + \sum_{t=2}^{T-1}\left(\frac{1}{1-\mu_x} - \frac{\mu_x}{1-\mu_x}\right)\mathbb{E}[F(x^{(t)})] - TF(x_*)$$

$$= F(x^{(1)}) - \frac{\mu_x}{1-\mu_x}F(x^{(1)}) + \frac{1}{1-\mu_x}\mathbb{E}[F(x^{(T)})] + \sum_{t=2}^{T-1}\left(\frac{1}{1-\mu_x} - \frac{\mu_x}{1-\mu_x}\right)\mathbb{E}[F(x^{(t)})] - TF(x_*)$$

$$= -\frac{\mu_x}{1-\mu_x}F(x^{(1)}) + \frac{\mu_x}{1-\mu_x}\mathbb{E}[F(x^{(T)})] + \sum_{t=1}^{T}\mathbb{E}[F(x^{(t)})] - TF(x_*).$$

Plugging-in this simplified expression, moving the extra $F(x^{(1)}) - F(x^{(t)})$ term to the right-hand side, and simplifying gives:

$$\sum_{t=1}^{T}\mathbb{E}\left[F(x^{(t)}) - F(x_*)\right] \le \sum_{t=1}^{T}\mathbb{E}[\langle\nabla F(y^{(t)}), z^{(t)} - x_*\rangle] + \frac{\mu_x}{1-\mu_x}\mathbb{E}\left[F(x^{(1)}) - F(x^{(T)})\right]$$

$$- \frac{1}{1-\mu_y}\sum_{t=1}^{T}\mathbb{E}[B_F(y^{(t)}, x^{(t)})] - \frac{\mu_y}{1-\mu_y}\sum_{t=1}^{T}\mathbb{E}[B_F(x^{(t)}, y^{(t)})]$$

$$- \frac{\mu_x}{1-\mu_x}\sum_{t=1}^{T}\mathbb{E}[B_F(x_{t-1}, x^{(t)})].$$

We get a bound on the average iterate $\bar{x}_T = \sum_{t=1}^{T} x^{(t)}$ by dividing by $T$ and applying Jensen's inequality:

$$\mathbb{E}[F(\bar{x}_T) - F(x_*)] \leq \frac{1}{T}\mathbb{E}\sum_{t=1}^{T}\langle\nabla F(y^{(t)}), z^{(t)} - x_*\rangle + \frac{\mu_x}{1 - \mu_x}\frac{1}{T}\mathbb{E}\left[F(x^{(1)}) - F(x^{(T)})\right]$$

$$- \frac{1}{1 - \mu_y}\frac{1}{T}\mathbb{E}\sum_{t=1}^{T}B_F(y^{(t)}, x^{(t)}) - \frac{\mu_y}{1 - \mu_y}\frac{1}{T}\mathbb{E}\sum_{t=1}^{T}B_F(x^{(t)}, y^{(t)})$$

$$- \frac{\mu_x}{1 - \mu_x}\frac{1}{T}\mathbb{E}\sum_{t=1}^{T}B_F(x_{t-1}, x^{(t)}).$$

Finally, we use $F(x_*) \leq F(x^{(T)})$ to get the claimed bound. $\qquad\square$

**Corollary 2.** *Assume that the base optimizer has regret guarantees $\sum_{t=1}^{T}\mathbb{E}[\langle\nabla F(y^{(t)}), z^{(t)} - x_*\rangle] = \mathcal{O}(\sqrt{T})$. Then:*

$$\mathbb{E}[F(\bar{x}^{(T)}) - F(x_*)] = \mathcal{O}\left(\frac{1}{\sqrt{T}}\right).$$

*Proof.* Note that we can upper bound the inequality in Theorem 1 by ignoring the negative Bregman divergence terms, i.e.,

$$\mathbb{E}[F(\bar{x}^{(T)}) - F(x_*)] \leq \frac{1}{T}\sum_{t=1}^{T}\mathbb{E}[\langle\nabla F(y^{(t)}), z^{(t)} - x_*\rangle] + \frac{\mu_x}{1 - \mu_x}\frac{1}{T}\mathbb{E}\left[F(x^{(1)}) - F(x_*)\right].$$

The result follows from noting that the first term is $\mathcal{O}(1/\sqrt{T})$ and the second term is $\mathcal{O}(1/T)$. $\quad\square$

## E  EXPERIMENTAL DETAILS

### E.1  COMPARISON BETWEEN GPA AND NESTEROV

In order to validate that DiLoCo's performance can only be matched or improved upon with decoupled interpolation constants in GPA, we test the case where $\mu_x = \mu_y$, which corresponds to Nesterov's primal averaging formulation in equation 4. Here, we apply the same heuristic for $\mu_x = \mu^{1/H}$ and also to $\mu_y$. We show the behavior for one particular choice of learning rate $3 \cdot 10^{-3}$, but observe that the same conclusions can be drawn for other choices as well. This is closely related to non-distributed DiLoCo with a single inner step.

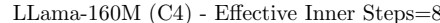

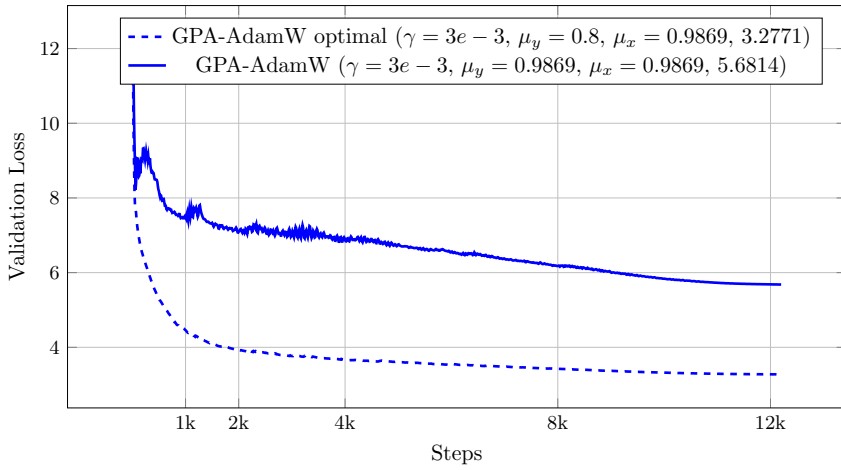

Figure 5: Comparison between Nesterov's primal averaging formulation with coupled constants $\mu_x = \mu_y$ and GPA with decoupled constants.

In Figure 5, we observe that coupling the interpolation constants is sub-optimal, and decoupling these coefficients is indeed necessary for optimal performance from GPA.

## E.2 Additional Validation Loss Curves for Different Effective Number of Inner Steps

In Figures 6 and 7, we provide additional validation loss curves for the cases where the effective number of inner steps equals 8 or 32, respectively. The results are generally consistent with the case where the number of inner steps is equal to 16 in Figure 3. When the effective number of inner steps is 32, we observe that AdamW outperforms DiLoCo for approximately the first 2,000 steps.

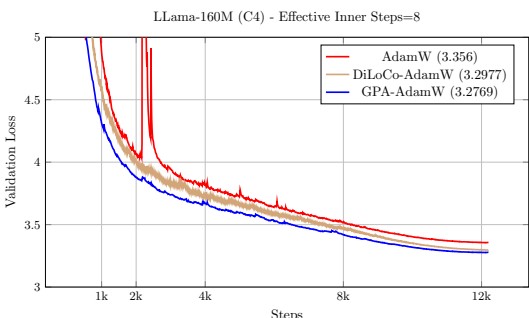

Figure 6: Validation loss versus steps for GPA, DiLoCo and AdamW when the effective number of inner steps equals 8.

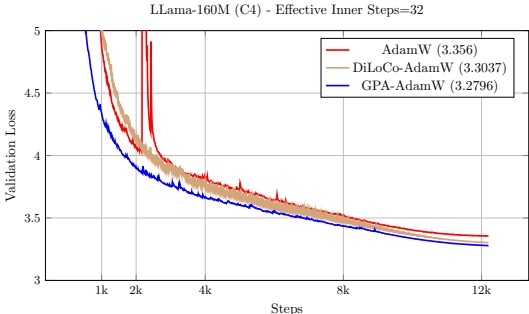

Figure 7: Validation loss versus steps for GPA, DiLoCo and AdamW when the effective number of inner steps equals 32.

## E.3 Hyperparameter Sweeps for Llama-160M

**Training setup.** We evaluate AdamW, DiLoCo-AdamW, and GPA-AdamW by pre-training the 160 million parameter Llama 3 model on the C4 dataset from scratch (Raffel et al., 2019). We follow the Chinchilla-optimal token budget of roughly 3.2 billion tokens (Hoffmann et al., 2022). All of our experiments are conducted on a single machine equipped with eight H100 GPUs (97GB memory). We use a batch size of 128 sequences with a sequence length of 2048 tokens, resulting in a total batch size of about 262,144 tokens. A summary of the hyperparameter sweeps are provided in Table 2.

**Hyperparameter tuning strategy.**

- For AdamW, we fix $(\beta_1, \beta_2) = (0.9, 0.999)$ and $\epsilon = 10^{-8}$, and sweep the learning rate from $5 \cdot 10^{-5}$ through $3 \cdot 10^{-3}$.
- For DiLoCo-AdamW, we fix the inner optimizer's hyperparameters to AdamW's optimal hyperparameters, and sweep the outer learning rate from $[0.25, 1.0]$ and the outer momentum from $[0.7, 0.99]$. We also sweep through the number of inner steps from $[1, 128]$ with powers of 2.

- For GPA-AdamW, we use the optimal AdamW hyperparameters, and sweep $\mu_x$ based on the number of inner steps in DiLoCo (see Section 3.1). We sweep $\mu_y$ over a fine granular range from $[0.8, 0.999]$. We also increased the learning rate when possible.

All runs use a learning rate schedule that applies linear warmup through the initial 10% of training, then cosine decay through the rest of training to $1\%$ of the specified learning rate. By default, we apply gradient clipping, with a clipping factor of $1.0$; weight decay is also fixed to $0.1$. A summary of the hyperparameter sweeps are provided in Table 2 in Appendix E.

**Summary of hyperparameter sweeps.** We summarize the hyperparameter sweeps used in our experiments in Table 2. In Table 3, we provide a table of conversions from optimal choices of $\mu$ and $H$ in DiLoCo to GPA's choice of $\mu_x$.

Table 2: Summary of hyperparameter sweeps used in the experiments.

| Hyperparameter | AdamW | DiLoCo-AdamW | GPA-AdamW |
|---|---|---|---|
| Batch size | 262K tokens | 262K tokens | 262K tokens |
| Sequence length | 2048 | 2048 | 2048 |
| Weight decay | 0.1 | 0.1 | 0.1 |
| Total training tokens | 3.2B | 3.2B | 3.2B |
| Total training steps | 12208 | 12208 | 12208 |
| Inner optimizer | AdamW | AdamW | GPA-AdamW |
| Inner optimizer lr | 5e-5, 1e-4, 2e-4, 3e-4, 5e-4, 7e-4, 1e-3, 3e-3 | 5e-4, 7e-4, 1e-3, 3e-3, 5e-3, 8e-3, 1e-2, 3e-2 | 5e-4, 7e-4, 1e-3, 3e-3, 5e-3, 8e-3, 1e-2, 3e-2 |
| Inner Adam $\beta_1$ | 0.9 | 0.9 | 0.5, 0.7, 0.9 |
| Inner Adam $\beta_2$ | 0.999 | 0.999 | 0.999 |
| Inner Adam $\epsilon$ | $10^{-8}$ | $10^{-8}$ | $10^{-8}$ |
| Warmup fraction | 10% | 10% | 10% |
| Learning rate schedule | cosine | cosine | cosine |
| Learning rate min fraction % | 0.01 | 0.01 | 0.01 |
| GPA coeff $\mu_y$ | - | - | 0.8, 0.9, 0.95, 0.9740, 0.9869, 0.99, 0.9913, 0.9934, 0.9956, 0.9967, 0.9978, 0.9984, 0.9989, 0.9992 |
| GPA coeff $\mu_x$ | - | - | 0.9, 0.9740, 0.9869, 0.9934, 0.9967, 0.9984, 0.9992 |
| Outer optimizer | - | Nesterov | - |
| Outer lr | - | 0.25, 0.5, 0.75, 1.0 | - |
| Outer momentum | - | 0.7, 0.9, 0.95, 0.9913, 0.9967, 0.9984, 0.9989, 0.9992 | - |
| Communication frequency $H$ | - | 1, 8, 16, 32, 64, 128 | - |

Table 3: Correspondence between the number of inner steps $H$ and momentum coefficient $\mu_{\text{diloco}}$ in DiLoCo and the momentum coefficient $\mu_x$ in GPA. The values of $\mu_x$ were computed using the expression $\mu_x = \mu_{\text{diloco}}^{1/H}$, with $\mu_{\text{diloco}} = 0.9$ and $H$ as the number of inner steps.

| Number of inner steps (DiLoCo) | $\mu_x$ (GPA) |
|---|---|
| 1 | 0.9000 |
| 4 | 0.9740 |
| 8 | 0.9869 |
| 16 | 0.9934 |
| 32 | 0.9967 |
| 64 | 0.9984 |
| 128 | 0.9992 |

### E.4 HYPERPARAMETER SWEEPS FOR LLAMA-1B

**Training setup.** We use the same dataset as in the smaller Llama model, but train longer for 50 billion tokens. To incorporate the larger workload, we utilize two machines (total of 16 H100 GPUs)

for each experiment, with an increased global batch size of 256 sequences with a sequence length of 2048 tokens, resulting in a total batch size of about 524,288 tokens.

**Hyperparameter tuning strategy.**

- For AdamW, we fix $(\beta_1, \beta_2) = (0.975, 0.95)$ since these were found to be the optimal values for this model following a sweep across a wide grid. We set $\epsilon = 10^{-8}$, and sweep the learning rate from $3 \cdot 10^{-4}$ through $8 \cdot 10^{-3}$.

- For DiLoCo-AdamW, we tested two sets of beta values: the tuned configuration used by the AdamW baseline $(\beta_1, \beta_2) = (0.975, 0.95)$ and another commonly used default from the recent work on DiLoCo $(\beta_1, \beta_2) = (0.9, 0.95)$ (Kallusky et al., 2025). The rest of the AdamW hyperparameters remain the same as the AdamW baseline. We sweep the outer learning rate in $\{0.75, 0.95\}$ and the outer momentum in $\{0.25, 0.7, 0.9\}$. We tuned the learning rate in $\{3 \cdot 10^{-4}, 8 \cdot 10^{-4}\}$. (We found even larger learning rates to be unstable for DiLoCo.) We also sweep through the number of inner steps in $\{8, 16, 32, 64, 128\}$.

- For GPA-AdamW, we provide the same two sets of beta values used for DiLoCo and keep the rest of the AdamW hyperparameter identical as the baselines. We sweep $\mu_x$ based on the number of inner steps in DiLoCo (see Table 3) corresponding to $\{8, 16, 32, 64, 128\}$. We tune $\mu_y$ in $\{0.8, 0.9\}$ since these were found to be more or less robust values based on several GPA runs. We tuned the learning rate in $\{3 \cdot 10^{-4}, 8 \cdot 10^{-4}, 1 \cdot 10^{-3}, 3 \cdot 10^{-3}, 5 \cdot 10^{-3}\}$.

### E.5 HYPERPARAMETER SWEEPS FOR VIT IMAGENET EXPERIMENTS

We pre-train the `vit_small_patch16_224.augreg_in21k` (ViT-S/16) model from `timm` on resolution 224, without fine-tuning it to the test resolution. We consider two settings based on the value of batch size: smaller batch size 4,096 and a larger value of 16,384. We train for 300 epochs in the smaller batch size regime, and for 200 epochs in the larger batch size regime. We tuned the methods separately in both settings, using the average over 2 random seeds to select the best parameters and then run the best-performing selection on 8 random seeds in total. For all methods, we used gradient clipping with norm 1, and warmed-up the learning rate linearly over the first 5 epochs and then decayed with cosine scheduler to $\times 0.001$ of the peak learning rate.

For data augmentations, we use RandAugment with strategy "rand-m15-n2", cutmix $\alpha = 1$, mixup with probability $0.5$ and $\alpha = 0.8$, no dropout, and no label smoothing. This setup has been reported to provide high validation accuracy values. For privacy reasons, we use the version of ImageNet-1k with faces blurred.

**Hyperparameter tuning strategy.**

- For AdamW, we fix $(\beta_1, \beta_2) = (0.9, 0.999)$ and $\epsilon = 10^{-8}$, which is standard for ImageNet training. We tuned learning rate across values $\{0.001, 0.003, 0.005, 0.007\}$ and weight decay across values $\{0.05, 0.1, 0.15, 0.2\}$.

- For GPA-AdamW, we fix $(\beta_1, \beta_2) = (0.8, 0.999)$ and $\epsilon = 10^{-8}$. We tuned weight decay and learning across the same values as for AdamW. We tested values of $\mu_y$ from $\{0.1, 0.2, 0.3, 0.5, 0.8, 0.9\}$. While the difference between them is less than $0.5\%$ validation accuracy, we found $\mu_y = 0.8$ to give the best results on 16,384 batch size runs and $\mu_y = 0.1$ to give the best results on 4,096 batch size.

The optimal learning rate and weight decay values were equal $0.005$ and $0.1$ for both methods in both settings.

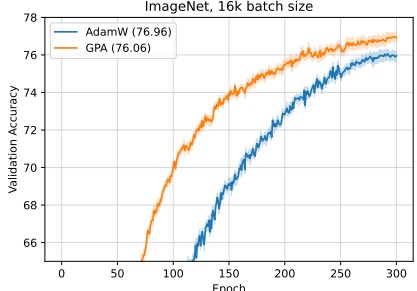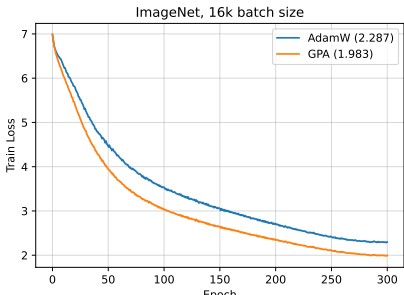

Figure 8: Comparison of AdamW and GPA on ImageNet ViT-S/16 from `timm` with data augmentations with a 16,384 batch size.

