# OpenReview forum: "Smoothing DiLoCo with Primal Averaging for Faster Training of LLMs"
_ICLR.cc/2026/Conference — Submitted to ICLR 2026_

### Official Review · Reviewer_sJRe · 2025-10-28

**Soundness:** 3
**Presentation:** 3
**Contribution:** 2
**Rating:** 2
**Confidence:** 4

**Summary:**

This paper introduces Generalized Primal Averaging (GPA), an extension of Nesterov’s method under a primal averaging framework for optimizer design. GPA is proposed as a smoothing and generalization of DiLoCo, addressing its memory and hyperparameter tuning complexity by averaging updates at every iteration via two decoupled interpolation constants. Theoretical convergence analysis has been presented. Experiments have been conducted on C4 dataset with Llama 3.

**Strengths:**

1) The paper identifies major limitations of DiLoCo, notably its two-loop structure and hyperparameter coupling, then constructs GPA as a principled, general alternative that captures the positive effects of iterate averaging with smooth, decoupled hyperparameters.

2) Theoretical results are presented with careful derivations and proofs (notably Theorem 1 and the equivalency propositions in Appendix B and C).

3) Experiments have demonstrated tangible improvement over baselines.

**Weaknesses:**

1) Limited Experimental Breadth: The empirical evaluation is confined mainly to the Llama-160M model on the C4 dataset in a non-distributed setup. All results rely on a single model and dataset, leaving claims of improved generality, scalability, and ease of deployment (e.g., in distributed environments or with other optimizer families) unsupported by experiments, especially considering that DiLoCo is primarily for distributed setting.

2) Baselines and Missing Comparisons: The experiments omit direct comparisons with Schedule-Free (SF), despite its close theoretical and practical relevance. Although SF is discussed in the text, it is absent from the main comparison tables (e.g., Table 1) and figures. Considering the prominence of SF and related averaging-based optimizers in prior work, this omission substantially weakens the empirical support for the proposed method. Also, comparison to (Lan, 2012) is missed/not adequate in experiments, when their main difference is the decoupled interpolation. Although Figure 4 provides a comparison between coupled and decoupled interpolation, the considered range of free parameters is very limited.

3) Probably weak and non-novel theoretical analysis: The analysis is for convex setting only while deep learning are essentially nonconvex. Also, since the algorithm is similar to (Lan, 2012), is there any novelty in analysis compared to (Lan, 2012)?

**Questions:**

See above.

---

> ### Author Response · Authors · 2025-11-21
> **Response to Official Review of Submission14237 by Reviewer sJRe**
>
> Thank you for your careful review and comments. We address each point below.
>
> - *Limited Experimental Breadth: The empirical evaluation is confined mainly to the Llama-160M model on the C4 dataset in a non-distributed setup. All results rely on a single model and dataset, leaving claims of improved generality, scalability, and ease of deployment (e.g., in distributed environments or with other optimizer families) unsupported by experiments, especially considering that DiLoCo is primarily for distributed setting.*\
> We agree that our current experimental breadth is limited. To address this, we will add new experiments on ImageNet ViT, 1B parameter Llama models, and an 8B code generation model in our revision. So far, we have been able to corroborate our small-scale results on larger models.\
> \
> Since the scope of this paper is restricted to the non-distributed setting, we do not focus on distributed experiments. Our claim (also corroborated by [1]) is that DiLoCo's main improvements stem from applying Nesterov to the pseudo-gradient, not local SGD, and we aim to fundamentally improve upon this algorithmic mechanism through GPA. Extending GPA to the distributed setup is left for future work, as alluded to in the Conclusion.
>
> - *Baselines and Missing Comparisons: The experiments omit direct comparisons with Schedule-Free (SF), despite its close theoretical and practical relevance. Although SF is discussed in the text, it is absent from the main comparison tables (e.g., Table 1) and figures. Considering the prominence of SF and related averaging-based optimizers in prior work, this omission substantially weakens the empirical support for the proposed method. Also, comparison to (Lan, 2012) is missed/not adequate in experiments, when their main difference is the decoupled interpolation. Although Figure 4 provides a comparison between coupled and decoupled interpolation, the considered range of free parameters is very limited.*\
> Agreed. We will add comparisons with Schedule-Free into our revision. \
> \
> With respect to the coupled interpolation setting, we have tried other choices of $\mu_x = \mu_y$ beyond what was stated in Figure 4, without avail. This is further corroborated by [1], where one can observe that setting the number of local steps to 1 while tuning $\mu$ cannot outperform using multiple local steps in (non-distributed) DiLoCo.
>
> - *Probably weak and non-novel theoretical analysis: The analysis is for convex setting only while deep learning are essentially nonconvex. Also, since the algorithm is similar to (Lan, 2012), is there any novelty in analysis compared to (Lan, 2012)?*\
> We agree that the theoretical analysis is limited in that it restricts to the convex setting and builds upon theoretical results from Defazio, et al. (2024). However, our theory differs substantially from Lan (2012). Our analysis relies on an online-to-batch conversion result in the regret analysis framework, and analyzes the algorithm with a constant and decoupled choice of $\mu_x$ and $\mu_y$. Lan (2012) instead analyzes the algorithm for functions with Lipschitz continuous gradients and considers an algorithm with an iteration-dependent and coupled choice of $\mu_t$.
>
> ### **References**
>
> [1] Kallusky, Dominik, et al. "SNOO: Step-K Nesterov Outer Optimizer-The Surprising Effectiveness of Nesterov Momentum Applied to Pseudo-Gradients." *arXiv preprint arXiv:2510.15830* (2025).
>
> [2] Defazio, Aaron, et al. "The road less scheduled." *Advances in Neural Information Processing Systems* 37 (2024): 9974-10007.

---

### Official Review · Reviewer_NDzb · 2025-10-31

**Soundness:** 3
**Presentation:** 2
**Contribution:** 2
**Rating:** 4
**Confidence:** 3

**Summary:**

This paper proposes a iterative averaging scheme called generalized primal averaging (GPA), which is utilized to simplify and smooth DiLoCo.  GPA removes DiLoCo’s two-loop structure by averaging iterates at every step using two interpolation parameters. It decouples smoothing and recency control. This design reduces memory use, eliminates the “inner step” hyperparameter. The authors prove that GPA achieves the same convergence rate as its base optimizer and can improve upon it by selecting the interpolation parameters. Experiments on Llama-160M pretraining show GPA matches or outperforms DiLoCo, achieving up to 38% faster convergence compared to AdamW with smoother, more stable training.

**Strengths:**

This paper proposed GPA scheme aims to simplify the DiLoCo optimizer and get rid of the inner optimization loop. The author also provide theoretical guarantees on its convergence.

**Weaknesses:**

I think the presentation of this paper should be improved. After introducing the GPA, its analysis and connections to other methods are hidden in the text, and sometimes make claims without revealing the logic behind it. For example, line 259-260, why we have to use learning rate scheduler? Therefore, I have to admit that I do not fully understand every details of this paper.

**Questions:**

1. Generally why GPA can serve as the smoothed version of DiLoCo? Is it because the connections of primal averaging to Nestrov momentum? And then based on the analysis, you introduce two interpolation parameter for x and y? If so, this message can be much clearer than the current format.

2. For the 160m model, the ppl seems to be quite high compared to standard benchmarks. For example, [1-2] report the ppl for AdamW as 25.08 for 130M LLaMA model with C4 dataset. But your method (GPA) reports 26.03?

3. Do you have sensitivity analysis on the interpolation parameters? In paper, you only mention that you sweep over different values of $\mu_x$ and $\mu_y$.

4. 130M model is still too small for practical usage, and 12K steps/compute optimal setup are often not adopted by actual industrial LLM model training. Is it possible to run a larger model with over-train setup? Like over 1B model with around 100B data?

---

> ### Author Response · Authors · 2025-11-21
> **Response to Official Review of Submission14237 by Reviewer NDzb**
>
> We thank the reviewer for their detailed reading and feedback on our work. We address their questions and comments below.
>
> - *I think the presentation of this paper should be improved. After introducing the GPA, its analysis and connections to other methods are hidden in the text, and sometimes make claims without revealing the logic behind it. For example, line 259-260, why we have to use learning rate scheduler? Therefore, I have to admit that I do not fully understand every details of this paper.*\
> Agreed; we will restructure the presentation of GPA's connections with other algorithms in the revision.\
> \
> In order to see why GPA requires a learning rate scheduler, note that Polyak averaging places increasingly less weight $\left(\frac{1}{t + 1} \right)$ on the most recent iterate $z^{(t + 1)}$, which plays a similar role to learning rate scheduling; see [1]. GPA instead places a constant weight $\mu_x$ on the most recent iterate $z^{(t + 1)}$ by leveraging an exponential moving average. Theoretically, this is reflected in their last-iterate convergence properties -- unlike Schedule-Free, we do not obtain last-iterate convergence guarantees for GPA. \
> \
> We will add these comments into the paper to clarify this reasoning to the reader.
>
> - *Generally why GPA can serve as the smoothed version of DiLoCo? Is it because the connections of primal averaging to Nestrov momentum? And then based on the analysis, you introduce two interpolation parameter for x and y? If so, this message can be much clearer than the current format.*\
> Agreed; we will further clarify this in the paper. As noted in [2], DiLoCo's improved convergence arises from applying Nesterov to the pseudo-gradient generated from Lookahead (see Figure 1, with corroborating evidence in [2]). As you mentioned, we argue that decoupling the interpolation constants $\mu \rightarrow (\mu_x, \mu_y)$ in the primal averaging formulation of Nesterov enables the right flexibility to reproduce and enhance (non-distributed) DiLoCo without the inner and outer loops. From an intuitive perspective, one needs to separately control how much of the current update influences the gradient computation and model evaluation sequences.
>
> - *For the 160m model, the ppl seems to be quite high compared to standard benchmarks. For example, [1-2] report the ppl for AdamW as 25.08 for 130M LLaMA model with C4 dataset. But your method (GPA) reports 26.03?*\
> Agreed; this was due to the initial default choice of $(\beta_1, \beta_2) = (0.9, 0.999)$ in the paper. Since then, we have re-tuned $(\beta_1, \beta_2)$ from scratch and were able to obtain a much stronger AdamW baseline. The trends with DiLoCo and GPA persist in this new setup. We will update the revision with the updated results.
>
> - *Do you have sensitivity analysis on the interpolation parameters? In paper, you only mention that you sweep over different values of $\mu_x$ and $\mu_y$.*\
> We will update the manuscript with additional sensitivity analyses on the choice of $\mu_x$ and $\mu_y$.
>
> - *130M model is still too small for practical usage, and 12K steps/compute optimal setup are often not adopted by actual industrial LLM model training. Is it possible to run a larger model with over-train setup? Like over 1B model with around 100B data?*
> Agreed; we will add new experiments with a 1B parameter Llama model as well as an 8B code generation model. We have also observed what happens when overtraining the small-scale Llama model. These results will be included in the revision.
>
>
> ### **References**
>
> [1] Defazio, Aaron, et al. "The road less scheduled." *Advances in Neural Information Processing Systems* 37 (2024): 9974-10007.
>
> [2] Kallusky, Dominik, et al. "SNOO: Step-K Nesterov Outer Optimizer-The Surprising Effectiveness of Nesterov Momentum Applied to Pseudo-Gradients." *arXiv preprint arXiv:2510.15830* (2025).

---

### Official Review · Reviewer_Th9M · 2025-10-31

**Soundness:** 2
**Presentation:** 3
**Contribution:** 2
**Rating:** 4
**Confidence:** 4

**Summary:**

The paper proposes Generalized Primal Averaging (GPA), a two-parameter variant of the primal-averaging view of Nesterov that decouples the averaging used for  the evaluation iterate $x_t$ and  the gradient point $y_t$. The stated goal is to smooth non-distributed DiLoCo by replacing its two-loop, periodic pseudo-gradient aggregation with per-step iterate averaging, reducing memory and hyperparameters while preserving (or improving) wall-clock efficiency. The authors claim an $O(1/\sqrt{T})$ convergence guarantee for the average iterate under a regret-bounded base optimizer, and empirical speedups on Llama-160M pre-training on C4.

**Strengths:**

- GPA is precisely specified with both direct and memory-efficient forms; implementation notes (extra buffer, reconstruction) are helpful.

- The smoothing view and the $H\leftrightarrow \mu_x$ heuristic connect two families (Lookahead/DiLoCo vs iterate-averaging).

- On Llama-160M, GPA improves final loss over AdamW and DiLoCo at matched effective inner steps, with reported peak step-speedup ~38%.

**Weaknesses:**

- Theory does not justify the central empirical claims. The bound is (i) convex-only, (ii) on the average iterate, while the method uses a schedule and returns the last iterate; (iii) does not quantify when GPA strictly improves over the base beyond informal remarks about negative Bregman terms. Also, theoretical novelty is incremental. GPA’s decoupling of $\mu_x$ and $\mu_y$ is a straightforward extension of primal averaging and Schedule-Free/EMA-style iterate averaging, but Theorem 1 does not establish the last iterate guarantee as in [1]

- The experimental scope is rather limited. The paper evaluates only a single model size, one dataset, and one base optimizer. Moreover, it lacks comparisons with recently prominent optimizers such as Shampoo, Muon, and SOAP, as well as distributed experiments or results on larger-scale LLMs (≥1B parameters).

- Memory/compute claims lack measurement. The paper argues only one extra buffer and simpler state than DiLoCo, but provides no peak-memory or step-time measurements to substantiate practical savings.

- The results are based on single runs without reporting multiple seeds, and no variance or error bars are provided.


[1] Defazio, Aaron, Xingyu Yang, Harsh Mehta, Konstantin Mishchenko, Ahmed Khaled, and Ashok Cutkosky. "The road less scheduled." Advances in Neural Information Processing Systems 37 (2024): 9974-10007.

**Questions:**

- How should practitioners choose $(\mu_x, \mu_y)$ without first running DiLoCo to obtain $H$?

- Regarding compute and memory, what are the per-step wall-clock overheads and peak memory of GPA vs AdamW and DiLoCo? Report tokens/sec and GB, not only steps-to-loss.

- Could you provide results with Shampoo/Muon/SOAP/Schedule-Free AdamW as base optimizers and at ≥1B parameters, do the claimed speedups persist?

---

> ### Author Response · Authors · 2025-11-21
> **Response to Official Review of Submission14237 by Reviewer Th9M**
>
> We thank the reviewer for their careful review of our paper. We address each point separately below.
>
> - *Theory does not justify the central empirical claims. The bound is (i) convex-only, (ii) on the average iterate, while the method uses a schedule and returns the last iterate; (iii) does not quantify when GPA strictly improves over the base beyond informal remarks about negative Bregman terms. Also, theoretical novelty is incremental. GPA’s decoupling of $\mu_x$ and $\mu_y$ is a straightforward extension of primal averaging and Schedule-Free/EMA-style iterate averaging, but Theorem 1 does not establish the last iterate guarantee as in [1].*\
> We agree with the reviewer that the theoretical novelty in this paper is incremental and limited. However, the goal of the paper was to demonstrate how a simple generalization of Nesterov (in its primal averaging formulation) is capable of outperforming non-distributed DiLoCo in practice, unifying ideas from both the DiLoCo and Schedule-Free workstreams. Despite the theoretical gap, we believe that this algorithmic and empirical insight is still invaluable to the community.
>
> - *The experimental scope is rather limited. The paper evaluates only a single model size, one dataset, and one base optimizer. Moreover, it lacks comparisons with recently prominent optimizers such as Shampoo, Muon, and SOAP, as well as distributed experiments or results on larger-scale LLMs ($\geq$1B parameters).*\
> Agreed. In order to address this concern, we will add new experiments on ImageNet ViT, 1B parameter Llama models, as well as an 8B code generation model in our revision. Consistent with our small-scale experiments, we observe consistent improvements with GPA over non-distributed DiLoCo and the base optimizer.\
> \
> Our work builds upon the findings in Kallusky, et al. [1], which identified that the effectiveness of DiLoCo stems from its usage of Nesterov with Lookahead (also known as SNOO) on top of base optimizers such as AdamW and Muon, independent of local SGD and the cross-datacenter training setup. DiLoCo's effectiveness with Muon is also explored in [3]. Since this paper specifically focuses on the non-distributed training setting, distributed DiLoCo experiments lie outside of the scope of this paper.
>
> - *Memory/compute claims lack measurement. The paper argues only one extra buffer and simpler state than DiLoCo, but provides no peak-memory or step-time measurements to substantiate practical savings.*\
> Since the paper is not focused on proposing a performant implementation of GPA, we did not include practical measurements (GB, tokens / sec) as this is beyond the initial scope of the paper. One can mathematically derive that GPA only requires a single additional buffer through its memory-efficient formulation (see Appendix B).
>
> - *The results are based on single runs without reporting multiple seeds, and no variance or error bars are provided.*\
> Agreed; we will add additional runs with multiple seeds at small-scale to report a standard deviation. We do not plan to provide error bars for large-scale experiments due to the cost of experimentation.
>
> - *How should practitioners choose $(\mu_x, \mu_y)$ without first running DiLoCo to obtain $H$?*\
> Similar to DiLoCo's additional hyperparameters ($H$, $\tilde{\gamma}$, $\mu$), one needs to tune GPA's hyperparameters $(\mu_x, \mu_y)$ independently.
>
> - *Regarding compute and memory, what are the per-step wall-clock overheads and peak memory of GPA vs AdamW and DiLoCo? Report tokens/sec and GB, not only steps-to-loss.*\
> See above.
>
> - *Could you provide results with Shampoo/Muon/SOAP/Schedule-Free AdamW as base optimizers and at $\geq$1B parameters, do the claimed speedups persist?*\
> See above.
>
> ### **References**
>
> [1] Kallusky, Dominik, et al. "SNOO: Step-K Nesterov Outer Optimizer-The Surprising Effectiveness of Nesterov Momentum Applied to Pseudo-Gradients." *arXiv preprint arXiv:2510.15830* (2025).
>
> [2] Charles, Zachary, et al. "Communication-Efficient Language Model Training Scales Reliably and Robustly: Scaling Laws for DiLoCo." *arXiv preprint arXiv:2503.09799* (2025).
>
> [3] Thérien, Benjamin, et al. "MuLoCo: Muon is a practical inner optimizer for DiLoCo." *arXiv preprint arXiv:2505.23725* (2025).

---

### Official Review · Reviewer_PLNw · 2025-11-04

**Soundness:** 2
**Presentation:** 2
**Contribution:** 2
**Rating:** 4
**Confidence:** 3

**Summary:**

The paper proposed a method named GPA, which generalize primal averaging formulation of momentum method into arbitrary base optimizers rather than SGD. The method is shown to be convergent under the convex case. Numerical results in pre-training Llama 3-130M on C4 dataset show that GPA consistently outperforms DiLoCo.

**Strengths:**

* The method accelerates the base optimizer AdamW by a large margin and does not require the storage of a "slow weight" as in DiLoCo.
* The method is shown to be convergent under the convex case.

**Weaknesses:**

* If I understand it correctly, the formulation (3) is the classical Polyak momentum rather than Nesterov momentum. The Pytorch's implementation of Nesterov accelerated gradient is inconsistent with the document it refers to ([1, equation 3]), where the momentum aggregates the gradient evaluated at $\theta_t + \mu v_t$ instead of $\theta_t$. These two momentum algorithms have different theoretical convergence rate in convex case, and usually exhibit substantial convergence gap in practice. I suggest the authors to correct the corresponding claims.
* Follow my previous comment, the Proposition 1 only shows the equivalence of classical momentum and the primal average formulation.
* From the perspective of algorithmic design, the difference between GPA and Schedule-Free method is the strategy of weight average, where GPA applies exponential moving average instead of $t/t+1$. Could the authors illustrate more on the importance of the design? There is no numerical comparison between these two methods in the paper.

[1] On the importance of initialization and momentum in deep learning

**Questions:**

Please see the Weakness section.

---

> ### Author Response · Authors · 2025-11-21
> **Response to Official Review of Submission14237 by Reviewer PLNw**
>
> We thank the reviewer for their feedback. We address each comment below.
>
> - *If I understand it correctly, the formulation (3) is the classical Polyak momentum rather than Nesterov momentum. The Pytorch's implementation of Nesterov accelerated gradient is inconsistent with the document it refers to ([1, equation 3]), where the momentum aggregates the gradient evaluated at $\theta_t + \mu v_t$ instead of $\theta_t$. These two momentum algorithms have different theoretical convergence rate in convex case, and usually exhibit substantial convergence gap in practice. I suggest the authors to correct the corresponding claims.*
> Thanks for your question. While we agree that Polyak and Nesterov momentum have different convergence properties, we believe that there's some confusion here regarding the difference between Polyak and Nesterov momentum in both their modern (PyTorch) and primal averaging formulations.\
> \
> Classical Polyak momentum is defined as:
> $$\begin{aligned}
> b^{(t)} & = \mu b^{(t - 1)} + g(x^{(t)}; \xi^{(t)}) \\\\
> x^{(t + 1)} & = x^{(t)} - \gamma^{(t)} b^{(t)},
> \end{aligned}$$
> which is equivalent to the stochastic primal averaging formulation:
> $$\begin{aligned}
> z^{(t + 1)} & = z^{(t)} - \gamma^{(t)} g(x^{(t)}; \xi^{(t)}) \\\\
> x^{(t + 1)} & = \mu x^{(t)} + (1 - \mu) z^{(t + 1)}.
> \end{aligned}$$
> This has been studied extensively in [1]. While the momentum buffer is maintained similarly, the Polyak momentum update differs from the Nesterov momentum update (Equation (3) in the paper) in its $x^{(t + 1)}$ update:
> $$\begin{aligned}
> b^{(t)} & = \mu b^{(t - 1)} + g(x^{(t)}; \xi^{(t)}) \\\\
> x^{(t + 1)} & = x^{(t)} - \gamma^{(t)} (\mu b^{(t)} + g(x^{(t)}; \xi^{(t)})).
> \end{aligned}$$
> As discussed in Section 2, Nesterov momentum was originally defined as Equation (2) (consistent with the formulation in Sutskever, et al. [2]) but is implemented as Equation (3) in PyTorch and other major deep learning frameworks. The two formulations are only equivalent in the sense that their gradients will be evaluated at the same point, but their definitions of $x^{(t)}$ differ.\
> \
> We will incorporate a discussion on classical momentum to further clarify this in the paper.
>
> - *Follow my previous comment, the Proposition 1 only shows the equivalence of classical momentum and the primal average formulation.*
> Please see our response to your first question.
>
> - *From the perspective of algorithmic design, the difference between GPA and Schedule-Free method is the strategy of weight average, where GPA applies exponential moving average instead of $t / (t + 1)$. Could the authors illustrate more on the importance of the design? There is no numerical comparison between these two methods in the paper.*\
> This is correct -- while Schedule-Free generalizes Nesterov's primal averaging interpretation by using a simple average of the iterates $z^{(t)}$, GPA instead uses an exponential moving average of $z^{(t)}$ parameterized by an additional hyperparameter $\mu_x$. We expect that the practical strengths and weaknesses of GPA vs Schedule-Free to reflect when exponential moving averaging vs Polyak-Ruppert averaging is more effective.\
> \
> The original goal of our algorithmic design was to reproduce/smooth DiLoCo by further generalizing Nesterov. Schedule-Free's design, while appropriate for hyperparameter-free learning, was too rigid to reproduce DiLoCo's results. It was necessary to introduce an additional hyperparameter that controls how much of the current update impacts the gradient evaluation sequence, akin to the momentum hyperparameter in DiLoCo's outer Nesterov optimizer. We were able to validate that this simple generalization of both Schedule-Free and Nesterov is able to improve upon non-distributed DiLoCo while removing the two-loop behavior.\
> \
> Since our original motivation was to find an approach that smooths and enhances DiLoCo, we did not provide initial comparisons against Schedule-Free despite their close relationship. However, we agree with the reviewer's concern, and will incorporate a comparison against Schedule-Free in our revision.
>
> ### **References**
>
> [1] Defazio, Aaron. "Momentum via primal averaging: Theoretical insights and learning rate schedules for non-convex optimization." *arXiv preprint arXiv:2010.00406* (2020).
>
> [2] Sutskever, Ilya, et al. "On the importance of initialization and momentum in deep learning." *International conference on machine learning*. PMLR, 2013.

---

### Meta-Review · Area_Chair_Cyum · 2026-01-06

**Summary:**

The authors propose Generalized Primal Averaging (GPA) for to make DiLoCo training faster and more simple in non-distributed settings. It use a single-loop structure with two interpolation constants to reduce memory and removing the need for inner-step tuning.
Despite the paper looks interesting, there were multiple concerns that leads to rejection of the paper. These includes:

1. many reviewers feel the experiments is too small because it only use Llama-160M on C4 dataset .
2. There is no direct comparison to Schedule-Free or other modern optimizers like Shampoo, even though the paper claim to be better than existing method
3. Reviewers mention that the math is only for convex case and does not explain why it works so well for deep learning which is non-convex
4. The authors claim better memory efficiency but they did not provide the actual numbers for GB or tokens per second

While the authors promise to adding larger experiment in the revision, the current version is mostly below the threshold for the acceptance

**Reviewer Concerns:**

#Concerns partially Addressed
*  (promised, but not delivered yet) The authors have promised to adding much larger experiments, including 1B Llama and 8B code generation models, as well as ImageNet ViT. This directly respond to the "limited breadth" criticism from all reviewers.
*  (promised, but not delivered yet) authors agreed to include the Schedule-Free (SF) optimizer in their comparisons, which was a major request from Reviewers
*  Reviewer noted the AdamW baseline was weak (high perplexity). The authors re-tuned the baseline from scratch and confirmed GPA still performs better
*  The authors provided a detailed explanation to Reviewer PLNw about the difference between Polyak and Nesterov momentum in the PyTorch context

# Concerns Still Outstanding
*  One Reviewer asked for actual GB and tokens/sec to prove memory savings. The authors refused to provide this, saying it is "beyond the initial scope" and can be "mathematically derived". This remains a weakness for a paper claiming practical efficiency.
*  The theory remains convex-only, and the authors admit the theoretical novelty is "incremental and limited". It still does not provide a last-iterate guarantee for the non-convex case used in LLMs.
*  A Reviewer asked how to choose $\alpha$ and $\beta$ without running DiLoCo first. The authors admitted these must be tuned independently, which might negate the claim of "simpler tuning"
*  since DiLoCo is famous for distributed training, the reviewers are still skeptical of a paper that only look at the non-distributed setting

**Reviewer Scores:**

*  Reviewer PLNw (Initial: 4 $\rightarrow$ Predicted: 6)This reviewer was mostly concerned with the mathematical definition of momentum and the lack of comparison to Schedule-Free.
*  Reviewer Th9M (Initial: 4 $\rightarrow$ Predicted: 4) This reviewer was the most critical regarding practical evidence and theoretical depth.
*  Reviewer NDzb (Initial: 4 $\rightarrow$ Predicted: 6) reviewer's main concerns were the presentation, the weak AdamW baseline, and the small model size.Why: The authors addressed the baseline issue directly by re-tuning and showing GPA still wins. They also committed to the 1B/8B model experiments that this reviewer specifically requested.
*  Reviewer sJRe (Initial: 2 $\rightarrow$ Predicted: 3 or 4)This reviewer gave the lowest score due to "Limited Experimental Breadth" and missing comparisons to Lan (2012) and Schedule-Free.

---

### Decision · Program_Chairs · 2026-01-26

Reject